# CausalGAN: Learning Causal Implicit Generative Models with Adversarial Training

**Murat Kocaoglu**[*] **Christopher Snyder**[*] **Alexandros G. Dimakis,**
**Sriram Vishwanath**

Department of Electrical and Computer Engineering
The University of Texas at Austin
Austin, TX, USA
`mkocaoglu@utexas.edu,22csnyder@gmail.com,`
`dimakis@austin.utexas.edu,sriram@austin.utexas.edu`

## Abstract

We introduce causal implicit generative models (CiGMs): models that allow sampling from not only the true observational but also the true interventional distributions. We show that adversarial training can be used to learn a CiGM, if the generator architecture is structured based on a given causal graph. We consider the application of conditional and interventional sampling of face images with binary feature labels, such as *mustache, young*. We preserve the dependency structure between the labels with a given causal graph. We devise a two-stage procedure for learning a CiGM over the labels and the image. First we train a CiGM over the binary labels using a Wasserstein GAN where the generator neural network is consistent with the causal graph between the labels. Later, we combine this with a conditional GAN to generate images conditioned on the binary labels. We propose two new conditional GAN architectures: CausalGAN and CausalBEGAN. We show that the optimal generator of the CausalGAN, given the labels, samples from the image distributions conditioned on these labels. The conditional GAN combined with a trained CiGM for the labels is then a CiGM over the labels and the generated image. We show that the proposed architectures can be used to sample from observational and interventional image distributions, even for interventions which do not naturally occur in the dataset.

## 1 Introduction

An implicit generative model (Mohamed & Lakshminarayanan (2016)) is a mechanism that can sample from a probability distribution without an explicit parameterization of the likelihood. Generative adversarial networks (GANs) arguably provide one of the most successful ways to train implicit generative models. GANs are neural generative models that can be trained using backpropagation to sample from very high dimensional nonparametric distributions (Goodfellow et al. (2014)). A *generator* network models the sampling process through feedforward computation given a noise vector. The generator output is constrained and refined through feedback by a competitive adversary network, called the discriminator, that attempts to distinguish between the generated and real samples. The objective of the generator is to maximize the loss of the discriminator (convince the discriminator that it outputs samples from the real data distribution). GANs have shown tremendous success in generating samples from distributions such as image and video (Vondrick et al. (2016)).

An extension of GANs is to enable sampling from the class conditional data distributions by feeding class labels to the generator alongside the noise vectors. Various neural network architectures have been proposed for solving this problem (Mirza & Osindero (2014); Odena et al. (2016); Antipov et al.

---

[*]Equal contribution

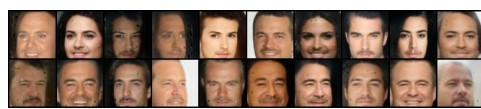

(a) Top: Intervened on Bald=1. Bottom: Conditioned on Bald = 1. $Male \rightarrow Bald$.

(b) Top: Intervened on Mustache=1. Bottom: Conditioned on Mustache = 1. $Male \rightarrow Mustache$.

Figure 1: Observational and interventional samples from CausalBEGAN. Our architecture can be used to sample not only from the joint distribution (conditioned on a label) but also from the interventional distribution, e.g., under the intervention do($Mustache = 1$). The two distributions are clearly different since $\mathbb{P}(Male = 1 | Mustache = 1) = 1$ and $\mathbb{P}(Bald = 1 | Male = 0) = 0$ in the data distribution $\mathbb{P}$.

(2017)). However, these architectures do not capture the dependence between the labels. Therefore, they do not have a mechanism to sample images *given a subset of the labels*, since they cannot sample the remaining labels. In this paper, we are interested in extending the previous work on conditional image generation by *i)* capturing the dependence between labels and *ii)* capturing *the causal effect* between labels. We can think of conditional image generation as a causal process: Labels determine the image distribution. The generator is a non-deterministic mapping from labels to images. This is consistent with the causal graph *"Labels cause the Image"*, denoted by $L \rightarrow I$, where $L$ is the random vector for labels and $I$ is the image random variable. Using a finer model, we can also include the causal graph between the labels, if available.

As an example, consider the causal graph between *Gender* ($G$) and *Mustache* ($M$) labels. The causal relation is clearly *Gender causes Mustache*, denoted by the graph $G \rightarrow M$. Conditioning on *Gender = male*, we expect to see males with or without mustaches, based on the fraction of males with mustaches in the population. When we condition on *Mustache = 1*, we expect to sample from males only since the population does not contain females with mustaches. In addition to sampling from conditional distributions, causal models allow us to sample from various different distributions called *interventional distributions*. An intervention is an experiment that fixes the value of a variable in a causal graph. This affects the distributions of the descendants of the intervened variable in the graph. But unlike conditioning, it does not affect the distribution of its ancestors. For the same causal graph, intervening on *Mustache = 1* would not change the distribution of *Gender*. Accordingly, the label combination (*Gender = female, Mustache = 1*) would appear as often as *Gender = female* after the intervention. Please see Figure 1 for some of our conditional and interventional samples, which illustrate this concept on the *Bald* and *Mustache* variables.

In this work we propose *causal implicit* generative models (CiGM): mechanisms that can sample not only from the correct joint probability distributions but also from the correct *conditional* and *interventional* probability distributions. Our objective is not to learn the causal graph: we assume that the true causal graph is given to us. We show that when the generator structure inherits its neural connections from the causal graph, GANs can be used to train causal implicit generative models. We use Wasserstein GAN (WGAN) (Arjovsky et al. (2017)) to train a CiGM for binary image labels, as the first step of a two-step procedure for training a CiGM for the images and image labels. For the second step, we propose two novel conditional GANs called CausalGAN and CausalBEGAN. We show that the optimal generator of CausalGAN can sample from the true conditional distributions (see Theorem 1).

We show that combining CausalGAN with a CiGM on the labels yields a CiGM on the labels and the image, which is formalized in Corollary 1 in Section 5. Our contributions are as follows:

- We observe that adversarial training can be used after structuring the generator architecture based on the causal graph to train a CiGM. We empirically show that WGAN can be used to learn a CiGM that outputs *essentially discrete*[1] labels, creating a CiGM for binary labels.
- We consider the problem of conditional and interventional sampling of images given a causal graph over binary labels. We propose a two-stage procedure to train a CiGM over the binary labels and the image. As part of this procedure, we propose a novel conditional GAN

---

[1] Each of the generated labels is sharply concentrated around 0 or 1 (Please see Figure 11a in the Appendix).

architecture and loss function. We show that the global optimal generator provably samples from the class conditional distributions.

- We propose a natural but nontrivial extension of BEGAN to accept labels: using the same motivations for margins as in BEGAN (Berthelot et al. (2017)), we arrive at a "margin of margins" term. We show empirically that this model, which we call CausalBEGAN, produces high quality images that capture the image labels.
- We evaluate our CiGM training framework on the labeled CelebA data (Liu et al. (2015)). We empirically show that CausalGAN and CausalBEGAN can produce label-consistent images *even for label combinations realized under interventions that never occur during training*, e.g., "woman with mustache"[2].

## 2 RELATED WORK

Using a GAN conditioned on the image labels has been proposed before: In Mirza & Osindero (2014), authors propose conditional GAN (CGAN): They extend generative adversarial networks to the setting where there is extra information, such as labels. Image labels are given to both the generator and the discriminator. In Odena et al. (2016), authors propose ACGAN: Instead of receiving the labels as input, the discriminator is now tasked with estimating the label. In Sricharan et al. (2017), the authors compare the performance of CGAN and ACGAN and propose an extension to the semi-supervised setting. In Chen et al. (2016), authors propose a new architecture called InfoGAN, which attempts to maximize a variational lower bound of mutual information between the inputs given to the generator and the image. To the best of our knowledge, the existing conditional GANs do not allow sampling from label combinations that do not appear in the dataset (Sricharan (2017)).

BiGAN (Donahue et al. (2017b)) and ALI (Dumoulin et al. (2017)) extend the standard GAN framework by also learning a mapping from the image space to a latent space. In CoGAN (Liu & Oncel (2016)) the authors learn a joint distribution over an image and its binary label by enforcing weight sharing between generators and discriminators. SD-GAN (Donahue et al. (2017a)) is a similar architecture which splits the latent space into "Identity" and "Observation" portions. To generate faces of the same person, one can then fix the identity portion of the latent code. If we consider the "Identity" and "Observation" codes to be the labels then SD-GAN can be seen as an extension of BEGAN to labels. This is, to the best of our knowledge, the only extension of BEGAN to accept labels before CausalBEGAN. It is not trivial to extend CoGAN and SD-GAN to more than two labels. Authors in Antipov et al. (2017) use CGAN of Mirza & Osindero (2014) with a one-hot encoded vector that encodes the age interval. A generator conditioned on this one-hot vector can then be used for changing the age attribute of a face image. Another application of generative models is in compressed sensing: Authors in Bora et al. (2017) give compressed sensing guarantees for recovering a vector, if the data lies close to the output of a trained generative model.

Using causal principles for deep learning and using deep learning techniques for causal inference has been recently gaining attention. In Lopez-Paz & Oquab (2016), the authors observe the connection between GAN layers, and structural equation models. Based on this observation, they use CGAN (Mirza & Osindero (2014)) to learn the causal direction between two variables from a dataset. In Lopez-Paz et al. (2017), the authors propose using a neural network in order to discover the causal relation between image class labels based on static images. In Bahadori et al. (2017), authors propose a new regularization for training a neural network, which they call causal regularization, in order to assure that the model is predictive in a causal sense. In a very recent work Besserve et al. (2017), authors point out the connection of GANs to causal generative models. However they see image as a cause of the neural net weights, and do not use labels. In an independent parallel work, authors in Goudet et al. (2017) propose using neural networks for learning causal graphs. Similar to us, they also use neural connections to mimic structural equations, but for learning the causal graph.

## 3 CAUSALITY BACKGROUND

In this section, we give a brief introduction to causality. Specifically, we use Pearl's framework (Pearl (2009)), i.e., structural causal models (SCMs), which uses structural equations and directed acyclic graphs between random variables to represent a causal model.

---

[2]This observation is not supported by theory since the distribution over the labels is not strictly positive.

Consider two random variables $X, Y$. Within the SCM framework and under the causal sufficiency assumption[3], $X$ *causes* $Y$ means that there exists a function $f$ and some unobserved random variable $E$, independent from $X$, such that *the value of $Y$ is determined based on the values of $X$ and $E$ through the function $f$*, i.e., $Y = f(X, E)$. Unobserved variables are also called *exogenous*. The causal graph that represents this relation is $X \to Y$. In general, a causal graph is a directed acyclic graph implied by the structural equations: The parents of a node $X_i$ in the causal graph, shown by $Pa_i$, represent the *causes* of that variable. The causal graph can be constructed from the structural equations as follows: The parents of a variable are those that appear in the structural equation that determines the value of that variable.

Formally, a structural causal model is a tuple $\mathcal{M} = (\mathcal{V}, \mathcal{E}, \mathcal{F}, \mathbb{P}_{\mathcal{E}}(.))$ that contains a set of functions $\mathcal{F} = \{f_1, f_2, \ldots, f_n\}$, a set of random variables $V = \{X_1, X_2, \ldots, X_n\}$, a set of exogenous random variables $\mathcal{E} = \{E_1, E_2, \ldots, E_n\}$, and a product probability distribution over the exogenous variables $\mathbb{P}_{\mathcal{E}}$. The set of observable variables $\mathcal{V}$ has a joint distribution implied by the distribution of $\mathcal{E}$, and the functional relations $\mathcal{F}$. The causal graph $D$ is then the directed acyclic graph on the nodes $\mathcal{V}$, such that a node $X_j$ is a parent of node $X_i$ if and only if $X_j$ is in the domain of $f_i$, i.e., $X_i = f_i(X_j, S, E_i)$, for some $S \subset V$. See the Appendix for more details.

An *intervention* is an operation that changes the underlying causal mechanism, hence the corresponding causal graph. An intervention on $X_i$ is denoted as $do(X_i = x_i)$. It is different from conditioning on $X_i$ in the following way: An intervention removes the connections of node $X_i$ to its parents, whereas conditioning does not change the causal graph from which data is sampled. The interpretation is that, for example, if we set the value of $X_i$ to 1, then it is no longer determined through the function $f_i(Pa_i, E_i)$. An intervention on a set of nodes is defined similarly. The joint distribution over the variables after an intervention (post-interventional distribution) can be calculated as follows: Since $D$ is a Bayesian network for the joint distribution, the observational distribution can be factorized as $\mathbb{P}(x_1, x_2, \ldots x_n) = \prod_{i \in [n]} \mathbb{P}(x_i | Pa_i)$, where the nodes in $Pa_i$ are assigned to the corresponding values in $\{x_i\}_{i \in [n]}$. After an intervention on a set of nodes $X_S \coloneqq \{X_i\}_{i \in S}$, i.e., $do(X_S = \mathbf{s})$, the post-interventional distribution is given by $\prod_{i \in [n] \setminus S} \mathbb{P}(x_i | Pa_i^S)$, where $Pa_i^S$ represents the following assignment: $X_j = x_j$ for $X_j \in Pa_i$ if $j \notin S$ and $X_j = \mathbf{s}(j)$ if $j \in S$[4].

In general it is not possible to identify the true causal graph for a set of variables without performing experiments or making additional assumptions. This is because there are multiple causal graphs that allow the same joint probability distribution even for two variables (Spirtes et al. (2001)). This paper does not address the problem of learning the causal graph: We assume that the causal graph is given to us, and we learn a causal model, i.e., the functions comprising the structural equations for some choice of exogenous variables[5]. There is significant prior work on learning causal graphs that could be used before our method (Spirtes et al. (2001); Heckerman (1995); Chickering (2002); Hoyer et al. (2008); Hyttinen et al. (2013); Hauser & Bühlmann (2014); Shanmugam et al. (2015); Lopez-Paz et al. (2015); Peters et al. (2016); Etesami & Kiyavash (2016); Quinn et al. (2015); Kocaoglu et al. (2017b;a)). When the true causal graph is unknown using a Bayesian network that respects the conditional independences in the data allows us to sample from the correct observational distributions. We explore the effect of the used Bayesian network in Section 8.10, 8.11.

## 4 CAUSAL IMPLICIT GENERATIVE MODELS

Implicit generative models can sample from the data distribution. However they do not provide the functionality to sample from interventional distributions. We propose *causal implicit generative models*, which provide a way to sample from both observational and interventional distributions.

We show that generative adversarial networks can also be used for training causal implicit generative models. Consider the simple causal graph $X \to Z \leftarrow Y$. Under the causal sufficiency assumption, this model can be written as $X = f_X(E_X), Y = f_Y(E_Y), Z = f_Z(X, Y, E_Z)$, where $f_X, f_Y, f_Z$ are some functions and $E_X, E_Y, E_Z$ are jointly independent variables. The following simple observation

---

[3]In a causally sufficient system, every unobserved variable affects not more than a single observed variable.

[4]With slight abuse of notation, we use $\mathbf{s}(j)$ to represent the value assigned to variable $X_j$ by the intervention rather than the $j$th coordinate of $\mathbf{s}$.

[5]Even when the causal graph is given, there will be many different sets of functions and exogenous noise distributions that explain the observed joint distribution for that causal graph. We are learning one such model.

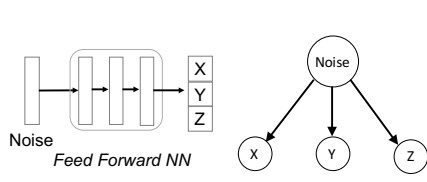
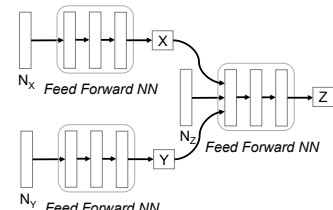

(a) Naive feedforward generator architecture and the causal graph it represents.

(b) Generator neural network architecture that represent the causal graph $X \rightarrow Z \leftarrow Y$.

Figure 2: (a) The causal graph implied by the naive feedforward generator architecture. (b) A neural network implementation of the causal graph $X \rightarrow Z \leftarrow Y$: Each feed forward neural net captures the function $f$ in the structural equation model $V = f(Pa_V, E)$.

is useful: *In the GAN training framework, generator neural network connections can be arranged to reflect the causal graph structure*. Please see Figure 2b for this architecture. The feedforward neural networks can be used to represent the functions $f_X, f_Y, f_Z$. The noise terms $(N_X, N_Y, N_Z)$ can be chosen as independent, complying with the condition that $(E_X, E_Y, E_Z)$ are jointly independent. Note that although we do not know the distributions of the exogenous variables, for a rich enough function class, we can use Gaussian distributed variables (Mooij et al. (2010)) $N_X, N_Y, N_Z$. Hence this feedforward neural network can be used to represents the causal models with graph $X \rightarrow Z \leftarrow Y$.

The following proposition is well known in the causality literature. It shows that given the true causal graph, two causal models that have the same observational distribution have the same interventional distributions for any intervention. $\mathbb{P}_V$ and $\mathbb{Q}_V$ stands for the distributions induced on the set of variables in $V$ by $\mathbb{P}_{N_1}$ and $\mathbb{Q}_{N_2}$, respectively.

**Proposition 1.** *Let $\mathcal{M}_1 = (D_1 = (V, E), N_1, \mathcal{F}_1, \mathbb{P}_{N_1}(.)), \mathcal{M}_2 = (D_2 = (V, E), N_2, \mathcal{F}_2, \mathbb{Q}_{N_2}(.))$ be two causal models, where $\mathbb{P}_{N_1}(.), \mathbb{Q}_{N_2}(.)$ are strictly positive densities. If $\mathbb{P}_V(.) = \mathbb{Q}_V(.)$, then $\mathbb{P}_V(.|do(S)) = \mathbb{Q}_V(.|do(S))$*

We have the following definition, which ties a feedforward neural network with a causal graph:

**Definition 1.** *Let $Z = \{Z_1, Z_2, \ldots, Z_m\}$ be a set of mutually independent random variables. A feedforward neural network $G$ that outputs the vector $G(Z) = [G_1(Z), G_2(Z), \ldots, G_n(Z)]$ is called **consistent** with a causal graph $D = ([n], E)$, if $\forall i \in [n]$, $\exists$ a set of feedforward layers $f_i$ such that $G_i(Z)$ can be written as $G_i(Z) = f_i(\{G_j(Z)\}_{j \in Pa_i}, Z_{S_i})$, where $Pa_i$ are the set of parents of $i$ in $D$, and $Z_{S_i} := \{Z_j : j \in S_i\}$ are collections of subsets of $Z$ such that $\{S_i : i \in [n]\}$ is a partition of $[m]$.*

Based on the definition, we can define causal implicit generative models as follows:

**Definition 2** (CiGM). *A feedforward neural network $G$ with output*

$$G(Z) = [G_1(Z), G_2(Z), \ldots, G_n(Z)], \tag{1}$$

*is called a causal implicit generative model for the causal model $\mathcal{M} = (D = ([n], E), N, \mathcal{F}, \mathbb{P}_N(.))$ if $G$ is consistent with the causal graph $D$ and $\mathbb{P}(G(Z) = \mathbf{x}) = \mathbb{P}_{[n]}(\mathbf{x}) > 0, \forall \mathbf{x}$.*

We propose using adversarial training where the generator neural network is consistent with the causal graph according to Definition 1, which is explained in the next section.

## 5 CAUSAL GENERATIVE ADVERSARIAL NETWORKS

CiGMs can be trained with samples from a joint distribution given the causal graph between the variables. However, for the application of image generation with binary labels, we found it difficult to simultaneously learn the joint label and image distribution[6]. For this application, we focus on

---

[6]Please see the Section 8.16 in the Appendix for our primitive result using this naive attempt.

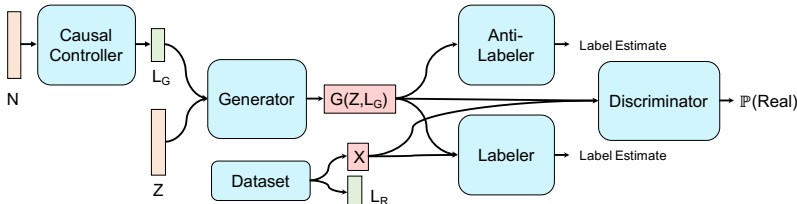

Figure 3: CausalGAN architecture: Causal controller is a pretrained causal implicit generative model for the image labels. Labeler is trained on the real data, Anti-Labeler is trained on generated data. Generator minimizes Labeler loss and maximizes Anti-Labeler loss.

dividing the task of learning a CiGM into two subtasks: First, we train a generative model over the labels, then train a generative model for the images conditioned on the labels. For this training to be consistent with the causal structure, we assume that the image node is always the sink node of the causal graph for image generation problems (Please see Figure 8 in Appendix). As we show next, our new architecture and loss function (CausalGAN) assures that the optimum generator outputs the label conditioned image distributions, under the assumption that the joint probability distribution over the labels is strictly positive[7]. Then for a strictly positive joint distribution between labels and the image, combining CiGM for only the labels with a label-conditioned image generator gives a CiGM for images and labels (see Corollary 1).

## 5.1 CAUSAL CONTROLLER

First we describe the adversarial training of a CiGM for binary labels. This generative model, which we call the *Causal Controller*, will be used for controlling which distribution the images will be sampled from when intervened or conditioned on a set of labels. As in Section 4, we structure the Causal Controller network to sequentially produce labels according to the causal graph. Since our theoretical results hold for binary labels, we prefer a generator which can sample from an essentially discrete label distribution[8]. However, the standard GAN training is not suited for learning a discrete distribution, since Jensen-Shannon divergence requires the support to be the same for giving meaningful gradients, which is harder with discrete data distributions. To be able to sample from a discrete distribution, we employ WGAN (Arjovsky et al. (2017)). We used the model of Gulrajani et al. (2017), where the Lipschitz constraint on the gradient is replaced by a penalty term in the loss.

## 5.2 CAUSALGAN

### 5.2.1 ARCHITECTURE

As part of the two-step process proposed in Section 4 for learning a CiGM over the labels *and* the image variables, we design a new conditional GAN architecture to generate the images based on the labels of the Causal Controller. Unlike previous work, our new architecture and loss function assures that the optimum generator outputs the label conditioned image distributions. We use a pretrained Causal Controller which is not further updated.

**Labeler and Anti-Labeler:** We have two separate labeler neural networks. *The Labeler* is trained to estimate the labels of images in the dataset. *The Anti-Labeler* is trained to estimate the labels of the images sampled from the generator, where image labels are those produced by the Causal Controller.

**Generator:** The objective of the generator is 3-fold: producing realistic images by competing with the discriminator, producing images consistent with the labels by minimizing the Labeler loss and avoiding unrealistic image distributions that are easy to label by maximizing the Anti-Labeler loss.

The most important distinction of CausalGAN with the existing conditional GAN architectures is that it uses an Anti-Labeler network in addition to a Labeler network. Notice that the theoretical guarantee

---

[7]This assumption does not hold in the CelebA dataset: $\mathbb{P}(Male = 0, Mustache = 1) = 0$. However, we will see that the trained model is able to extrapolate to these interventional distributions.

[8]Ignoring the theoretical considerations, adding noise to transform the labels artificially into continuous targets also works. However we observed better empirical convergence with this technique.

we develop in Section 5.2.3 does not hold without the Anti-Labeler. Intuitively, the Anti-Labeler loss discourages the generator network to output only few typical faces for a fixed label combination. This is a phenomenon that we call *label-conditioned mode collapse*. Minibatch-features are one of the most popular techniques used to avoid mode-collapse (Salimans et al. (2016)). However, the diversity within a batch of images due to different label combinations can make this approach ineffective for combating label-conditioned mode collapse. This effect is most prominent for rare label combinations. In general, using Anti-Labeler helps with faster convergence. Please see Section 9.4 in the Appendix for results.

### 5.2.2 Loss Functions

We present the results for a single binary label $l$. The results can be extended to more labels. For a single binary label $l$ and the image $x$, we use $\mathbb{P}_r(l, x)$ for the data distribution between the image and the labels. Similarly $\mathbb{P}_g(l, x)$ denotes the joint distribution between the labels given to the generator and the generated images. In our analysis we assume a perfect Causal Controller[9] and use the shorthand $\mathbb{P}_g(l = 1) = \mathbb{P}_r(l = 1) = \rho = 1 - \bar{\rho}$. Let $G(.), D(.), D_{LR}(.)$, and $D_{LG}(.)$ are the mappings due to generator, discriminator, Labeler, and Anti-Labeler respectively.

The generator loss function of CausalGAN contains label loss terms, the GAN loss in Goodfellow et al. (2014), and an added loss term due to the discriminator. With the addition of this term to the generator loss, we are able to prove that the optimal generator outputs the class conditional image distribution. This result is also true for multiple binary labels, which is shown in the Appendix.

For a fixed generator, Anti-Labeler solves the following optimization problem:

$$\max_{D_{LG}} \rho \mathbb{E}_{x \sim \mathbb{P}_g(x|l=1)} \left[\log(D_{LG}(x))\right] + \bar{\rho} \mathbb{E}_{x \sim \mathbb{P}_g(x|l=0)} \left[\log(1 - D_{LG}(x)\right]. \tag{2}$$

The Labeler solves the following optimization problem:

$$\max_{D_{LR}} \rho \mathbb{E}_{x \sim \mathbb{P}_r(x|l=1)} \left[\log(D_{LR}(x))\right] + \bar{\rho} \mathbb{E}_{x \sim \mathbb{P}_r(x|l=0)} \left[\log(1 - D_{LR}(x)\right]. \tag{3}$$

For a fixed generator, the discriminator solves the following optimization problem:

$$\max_{D} \mathbb{E}_{(l,x) \sim \mathbb{P}_r(l,x)} \left[\log(D(x))\right] + \mathbb{E}_{(l,x) \sim \mathbb{P}_g(l,x)} \left[\log(1 - D(x))\right]. \tag{4}$$

For a fixed discriminator, Labeler and Anti-Labeler, the generator solves the following problem:

$$\min_{G} \mathbb{E}_{(l,x) \sim \mathbb{P}_g(l,x)} \left[\log\left(\frac{1 - D(x)}{D(x)}\right)\right] - \rho \mathbb{E}_{x \sim \mathbb{P}_g(x|l=1)} \left[\log(D_{LR}(X))\right]$$
$$- \bar{\rho} \mathbb{E}_{x \sim \mathbb{P}_g(x|l=0)} \left[\log(1 - D_{LR}(X))\right] + \rho \mathbb{E}_{x \sim \mathbb{P}_g(x|l=1)} \left[\log(D_{LG}(X))\right]$$
$$+ \bar{\rho} \mathbb{E}_{x \sim \mathbb{P}_g(x|l=0)} \left[\log(1 - D_{LG}(X))\right]. \tag{5}$$

### 5.2.3 Theoretical Guarantees

We show that the best CausalGAN generator for the given loss function samples from the class conditional image distribution when Causal Controller samples from the true label distribution and the discriminator and labeler networks always operate at their optimum. We show this result for the case of a single binary label $l \in \{0, 1\}$. The proof can be extended to multiple binary variables, which is given in the Appendix. As far as we are aware of, this is the only conditional generative adversarial network architecture with this guarantee after CGAN[10].

First, we find the optimal discriminator for a fixed generator. Note that in (4), the terms that the discriminator can optimize are the same as the GAN loss in Goodfellow et al. (2014). Hence the optimal discriminator behaves the same. To characterize the optimum discriminator, labeler and anti-labeler, we have Proposition 2, Lemma 1 and Lemma 2 given in the appendix.

Let $C(G)$ be the generator loss for when the discriminator, Labeler and Anti-Labeler are at the optimum. Then the generator that minimizes $C(G)$ samples from the class conditional distributions:

---

[9]Even for multiple labels, we observe convergence in total variation distance. Please see Figure 11b.

[10]CGAN (Mirza & Osindero (2014)) can be shown to have the same guarantee. The difference of our architecture is that we do not feed image labels to the discriminator.

**Theorem 1.** *Assume $\mathbb{P}_g(l) = \mathbb{P}_r(l)$. Then the global minimum of the virtual training criterion $C(G)$ is achieved if and only if $\mathbb{P}_g(l, x) = \mathbb{P}_r(l, x)$, i.e., if and only if given a label $l$, generator output $G(z, l)$ has the same distribution as the class conditional image distribution $\mathbb{P}_r(x|l)$.*

Now we can show that our two stage procedure can be used to train a causal implicit generative model for any causal graph where the *Image* variable is a sink node, captured by the following corollary. $\mathcal{L}, \mathcal{I}, \mathcal{Z}_1, \mathcal{Z}_2$ represent the space of labels, images, and noise variables, respectively.

**Corollary 1.** *Suppose $C : \mathcal{Z}_1 \rightarrow \mathcal{L}$ is a causal implicit generative model for the causal graph $D = (\mathcal{V}, E)$ where $\mathcal{V}$ is the set of image labels and the observational joint distribution over these labels are strictly positive. Let $G : \mathcal{L} \times \mathcal{Z}_2 \rightarrow \mathcal{I}$ be a generator that can sample from the image distribution conditioned on the given label combination $L \in \mathcal{L}$. Then $G(C(Z_1), Z_2)$ is a causal implicit generative model for the causal graph $D' = (\mathcal{V} \cup \{Image\}, E \cup \{(V_1, Image), (V_2, Image), \dots (V_n, Image)\})$.*

In Theorem 1 we show that the optimum generator samples from the class conditional distributions given a single binary label. Our objective is to extend this result to the case with $d$ binary labels. First we show that if the Labeler and Anti-Labeler are trained to output $2^d$ scalars, each interpreted as the posterior probability of a particular label combination given the image, then the minimizer of $C(G)$ samples from the class conditional distributions *given $d$ labels*. This result is shown in Theorem 2 in the appendix. However, when $d$ is large, this architecture may be hard to implement. To resolve this, we propose an alternative architecture, which we implement for our experiments: We extend the single binary label setup and use cross entropy loss terms for each label. This requires Labeler and Anti-Labeler to have only $d$ outputs. However, although we need the generator to capture the joint label posterior given the image, this only assures that the generator captures each label's posterior distribution, i.e., $\mathbb{P}_r(l_i|x) = \mathbb{P}_g(l_i|x)$ (Proposition 3). This, in general, does not guarantee that the class conditional distributions will be true to the data distribution. However, for many joint distributions of practical interest, where *the set of labels are completely determined by the image*[11], we show that this guarantee implies that the joint label posterior will be true to the data distribution, implying that the optimum generator samples from the class conditional distributions. Please see Section 8.7 for the formal results and more details.

**Remark:** Note that the trained causal implicit generative models can also be used to sample from the counterfactual distributions if the exogenous noise terms are known. Counterfactual sampling require conditioning on an event and sampling from the push-forward of the posterior distributions of the exogenous noise terms under the interventional causal graph due to a possible intervention. This can be done through rejection sampling to observe the evidence, holding the exogenous noise terms consistent with the observed evidence and interventional sampling afterwards.

## 5.3 CAUSALBEGAN

In this section, we sketch a simple, but non-trivial extension of BEGAN where we feed image labels to the generator, leaving the details to the Appendix (Section 8.8). To accommodate interventional sampling, we again use the Causal Controller to produce labels.

In terms of architecture modifications, we use a Labeler network with a dual purpose: to label real images well and generated images poorly. This network can be seen as both analogous to the original two-roled BEGAN discriminator and analogous to the CausalGAN Labeler and Anti-Labeler.

In terms of margin modifications, we are motivated by the following observations: (1) Just as a better trained BEGAN discriminator creates more useful gradients for image quality, (2) a better trained Labeler is a prerequisite for meaningful gradients for label quality. Finally, (3) label gradients are most informative when the image quality is high. Each observation suggests a margin term; the final observation suggests a (necessary) *margin of margins* term comparing the first two margins.

## 6 RESULTS

In this section, we train CausalGAN and CausalBEGAN on the CelebA Causal Graph given in Figure 8. For this, we first trained the Causal Controller (See Section 8.11 for Causal Controller results.) on

---

[11]The dataset we are using arguably satisfies this condition.

the image labels of CelebA Causal Graph. Please see Section 9.2 for implementation details. The results are given in Figures 4, 5 for CausalGAN and Figures 6, 7 for CausalBEGAN. The difference between intervening and conditioning is clear through certain features. We implement conditioning through rejection sampling. See Naesseth et al. (2017); Graham & Storkey (2017) for other works on conditioning for implicit generative models.

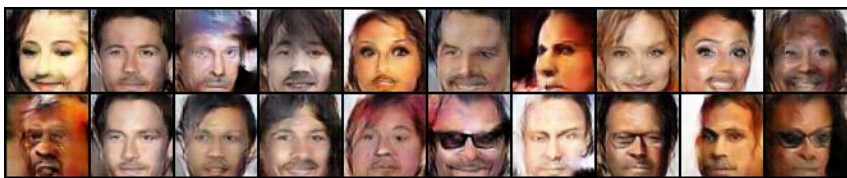

Top: Intervene Mustache=1, Bottom: Condition Mustache=1

Figure 4: Intervening/Conditioning on Mustache label in CelebA Causal Graph with CausalGAN. Since $Male \rightarrow Mustache$ in CelebA Causal Graph, we do not expect $do(Mustache = 1)$ to affect the probability of $Male = 1$, i.e., $\mathbb{P}(Male = 1|do(Mustache = 1)) = \mathbb{P}(Male = 1) = 0.42$. Accordingly, the top row shows both males and females with mustaches, even though the generator never sees the label combination $\{Male = 0, Mustache = 1\}$ during training. The bottom row of images sampled from the conditional distribution $\mathbb{P}(.|Mustache = 1)$ shows only male images.

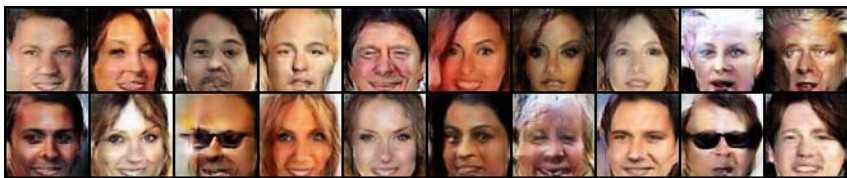

Top: Intervene Mouth Slightly Open=1, Bottom: Condition Mouth Slightly Open=1

Figure 5: Intervening/Conditioning on Mouth Slightly Open label in CelebA Causal Graph with CausalGAN. Since $Smiling \rightarrow MouthSlightlyOpen$ in CelebA Causal Graph, we do not expect $do(Mouth\ Slightly\ Open = 1)$ to affect the probability of $Smiling = 1$, i.e., $\mathbb{P}(Smiling = 1|do(Mouth\ Slightly\ Open = 1)) = \mathbb{P}(Smiling = 1) = 0.48$. However on the bottom row, conditioning on *Mouth Slightly Open* = 1 increases the proportion of smiling images (From $0.48$ to $0.76$ in the dataset), although 10 images may not be enough to show this difference statistically.

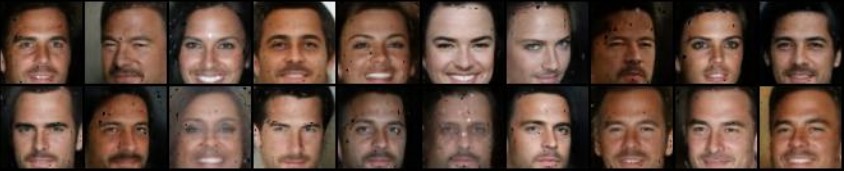

Top: Intervene Mustache=1, Bottom: Condition Mustache=1

Figure 6: Intervening/Conditioning on Mustache label in CelebA Causal Graph with CausalBEGAN. Since $Male \rightarrow Mustache$ in CelebA Causal Graph, we do not expect $do(Mustache = 1)$ to affect the probability of $Male = 1$, i.e., $\mathbb{P}(Male = 1|do(Mustache = 1)) = \mathbb{P}(Male = 1) = 0.42$. Accordingly, the top row shows both males and females with mustaches, even though the generator never sees the label combination $\{Male = 0, Mustache = 1\}$ during training. The bottom row of images sampled from the conditional distribution $\mathbb{P}(.|Mustache = 1)$ shows only male images.

## 7    CONCLUSION

We proposed a novel generative model with label inputs. In addition to being able to create samples *conditioned* on labels, our generative model can also sample from the *interventional* distributions. Our theoretical analysis provides provable guarantees about correct sampling under such interventions.

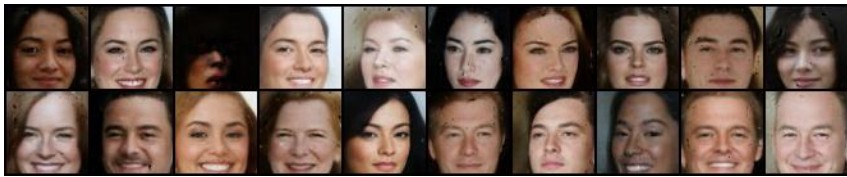

Top: Intervene Narrow Eyes=1, Bottom: Condition Narrow Eyes=1

Figure 7: Intervening/Conditioning on Narrow Eyes label in CelebA Causal Graph with CausalBEGAN. Since $Smiling \rightarrow Narrow\,Eyes$ in CelebA Causal Graph, we do not expect $do(Narrow\,Eyes = 1)$ to affect the probability of $Smiling = 1$, i.e., $\mathbb{P}(Smiling = 1 | do(Narrow\,Eyes = 1)) = \mathbb{P}(Smiling = 1) = 0.48$. However on the bottom row, conditioning on $Narrow\,Eyes = 1$ increases the proportion of smiling images (From $0.48$ to $0.59$ in the dataset), although 10 images may not be enough to show this difference statistically. As a rare artifact, in the dark image in the third column the generator appears to rule out the possibility of $Narrow\,Eyes = 0$ instead of demonstrating $Narrow\,Eyes = 1$.

Causality leads to generative models that are more creative since they can produce samples that are different from their training samples in multiple ways. We have illustrated this point for two models (CausalGAN and CausalBEGAN).

ACKNOWLEDGMENTS

We thank Ajil Jalal for the helpful discussions. This research has been supported by NSF Grants CCF, 1407278, 1422549, 1618689, 1564167, DMS 1723052, ARO YIP W911NF-14-1-0258, NVIDIA Corporation and ONR N000141512009.

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

## 8 APPENDIX

### 8.1 CAUSALITY BACKGROUND

Formally, a structural causal model is a tuple $\mathcal{M} = (\mathcal{V}, \mathcal{E}, \mathcal{F}, \mathbb{P}_E(.))$ that contains a set of functions $\mathcal{F} = \{f_1, f_2, \ldots, f_n\}$, a set of random variables $V = \{X_1, X_2, \ldots, X_n\}$, a set of exogenous random variables $\mathcal{E} = \{E_1, E_2, \ldots, E_n\}$, and a probability distribution over the exogenous variables $\mathbb{P}_\mathcal{E}$ [12]. The set of observable variables $\mathcal{V}$ has a joint distribution implied by the distributions of $\mathcal{E}$, and the functional relations $\mathcal{F}$. This distribution is the projection of $\mathbb{P}_\mathcal{E}$ onto the set of variables $\mathcal{V}$ and is shown by $\mathbb{P}_\mathcal{V}$. The causal graph $D$ is then the directed acyclic graph on the nodes $\mathcal{V}$, such that a node $X_j$ is a parent of node $X_i$ if and only if $X_j$ is in the domain of $f_i$, i.e., $X_i = f_i(X_j, S, E_i)$, for some $S \subset V$. The set of parents of variable $X_i$ is shown by $Pa_i$. $D$ is then a Bayesian network for the induced joint probability distribution over the observable variables $\mathcal{V}$. We assume causal sufficiency: Every exogenous variable is a direct parent of at most one observable variable.

### 8.2 PROOF OF PROPOSITION 1

Note that $D_1$ and $D_2$ are the same causal Bayesian networks Pearl (2009). Under the causal sufficiency assumption, interventional distributions for causal Bayesian networks can be directly calculated from the conditional probabilities and the causal graph. Thus, $\mathcal{M}_1$ and $\mathcal{M}_2$ have the same interventional distributions. $\qquad\square$

### 8.3 HELPER LEMMAS FOR CAUSALGAN

In this section we use $\mathbb{P}_r(l, x)$ for the joint data distribution over a single binary label $l$ and the image $x$. We use $\mathbb{P}_g(l, x)$ for the joint distribution over the binary label $l$ fed to the generator and the image $x$ produced by the generator. Later in Theorem 2, $l$ is generalized to be a vector.

The following restates Proposition 1 from Goodfellow et al. (2014) as it applies to our discriminator:

**Proposition 2** (Goodfellow et al. (2014)). *For fixed $G$, the optimal discriminator $D$ is given by*

$$D_G^*(x) = \frac{\mathbb{P}_r(x)}{\mathbb{P}_r(x) + \mathbb{P}_g(x)}. \tag{6}$$

Second, we identify the optimal Labeler and Anti-Labeler. We have the following lemma:

**Lemma 1.** *The optimum Labeler has $D_{LR}(x) = \mathbb{P}_r(l = 1|x)$.*

*Proof.* The proof follows the same lines as in the proof for the optimal discriminator. Consider the objective

$$\rho \mathbb{E}_{x \sim \mathbb{P}_r(x|l=1)} \left[\log(D_{LR}(x))\right] + (1 - \rho)\mathbb{E}_{x \sim \mathbb{P}_r(x|l=0)} \left[\log(1 - D_{LR}(x)\right]$$

$$= \int \rho \mathbb{P}_r(x|l = 1) \log(D_{LR}(x)) + (1 - \rho)\mathbb{P}_r(x|l = 0) \log(1 - D_{LR}(x))dx \tag{7}$$

Since $0 < D_{LR} < 1$, $D_{LR}$ that maximizes (3) is given by

$$D_{LR}^*(x) = \frac{\rho \mathbb{P}_r(x|l = 1)}{\mathbb{P}_r(x|l = 1)\rho + \mathbb{P}_r(x|l = 0)(1 - \rho)} = \frac{\rho \mathbb{P}_r(x|l = 1)}{\mathbb{P}_r(x)} = \mathbb{P}_r(l = 1|x) \tag{8}$$

$$\square \qquad\qquad\qquad\qquad\qquad\qquad\qquad\square$$

Similarly, we have the corresponding lemma for Anti-Labeler:

**Lemma 2.** *For a fixed generator with $x \sim \mathbb{P}_g(x)$, the optimum Anti-Labeler has $D_{LG}(x) = \mathbb{P}_g(l = 1|x)$.*

*Proof.* Proof is the same as the proof of Lemma 1. $\qquad\square$

---

[12]The definition provided here assumes causal sufficiency, i.e., there are no exogenous variables that affect more than one observable variable. Under causal sufficiency, Pearl's model assumes that the distribution over the exogenous variables is a product distribution, i.e., exogenous variables are mutually independent.

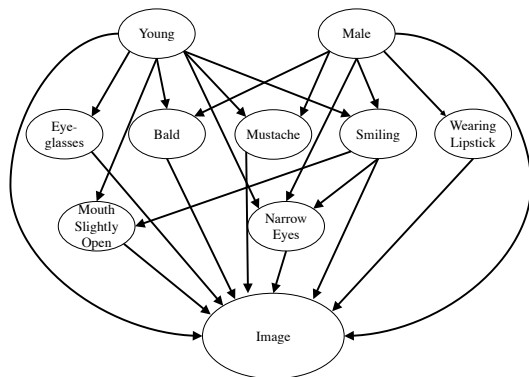

Figure 8: The causal graph used for simulations for both CausalGAN and CausalBEGAN, called CelebA Causal Graph (G1). We also add edges (see Appendix Section 8.10) to form the complete graph "cG1". We also make use of the graph rcG1, which is obtained by reversing the direction of every edge in cG1.

## 8.4 PROOF OF THEOREM 1

**Theorem 1.**
*Define $C(G)$ as the generator loss for when discriminator, Labeler and Anti-Labeler are at their optimum. Assume $\mathbb{P}_g(l) = \mathbb{P}_r(l)$, i.e., the Causal Controller samples from the true label distribution. Then the global minimum of the virtual training criterion $C(G)$ is achieved if and only if $\mathbb{P}_g(l, x) = \mathbb{P}_r(l, x)$, i.e., if and only if given a label $l$, generator output $G(z, l)$ has the same distribution as the class conditional image distribution $\mathbb{P}_r(x|l)$.*

*Proof.* For a fixed generator, the optimum Labeler $D_{LR}^*$, Anti-Labeler $D_{LG}^*$, and discriminator $D^*$ obey the following relations by Prop 2, Lemma 1, and Lemma 2:

$$(1 - D^*(x))/D^*(x) = \mathbb{P}_g(x)/\mathbb{P}_r(x)$$
$$D_{LR}^*(x) = \mathbb{P}_r(l = 1|x)$$
$$D_{LG}^*(x) = \mathbb{P}_g(l = 1|x).$$

(9)

Then substitution into the generator objective in (5) yields

$$C(G) = \mathbb{E}_{x \sim p_g(x)} \left[ \log \left( \frac{1 - D^*(x)}{D^*(x)} \right) \right] - \rho \mathbb{E}_{x \sim p_g^1(x)} \left[ \log(D_{LR}^*(X)) \right] - \bar{\rho} \mathbb{E}_{x \sim p_g^0(x)} \left[ \log(1 - D_{LR}^*(X)) \right]$$

$$+ \rho \mathbb{E}_{x \sim p_g^1(x)} \left[ \log(D_{LG}^*(X)) \right] + \bar{\rho} \mathbb{E}_{x \sim p_g^0(x)} \left[ \log(1 - D_{LG}^*(X)) \right]$$

$$= \mathbb{E}_{x \sim p_g(x)} \left[ \log \left( \frac{\mathbb{P}_g(x)}{\mathbb{P}_r(x)} \right) \right] - \mathbb{E}_{(l,x) \sim \mathbb{P}_g(l,x)} \left[ \log(\mathbb{P}_r(l|x)) \right] + \mathbb{E}_{(l,x) \sim \mathbb{P}_g(l,x)} \left[ \log(\mathbb{P}_g(l|x)) \right]$$

(10)

$$= \mathbb{E}_{(l,x) \sim \mathbb{P}_g(l,x)} \left[ \log \left( \frac{\mathbb{P}_g(x)}{\mathbb{P}_r(x)} \right) + \log(\mathbb{P}_g(l|x)) - \log(\mathbb{P}_r(l|x)) \right]$$

$$= \mathbb{E}_{(l,x) \sim \mathbb{P}_g(l,x)} \left[ \log \left( \frac{\mathbb{P}_g(l, x)}{\mathbb{P}_d(l, x)} \right) \right]$$

$$= KL(\mathbb{P}_g \parallel \mathbb{P}_d).$$

(11)

where KL is the Kullback-Leibler divergence, which is minimized if and only if $\mathbb{P}_g = \mathbb{P}_d$ jointly over labels and images. (10) is due to the fact that $\mathbb{P}_r(l = 1) = \mathbb{P}_g(l = 1) = \rho$. $\qquad\square$

## 8.5 PROOF OF COROLLARY 1

**Corollary 1.** *Suppose $C : \mathcal{Z}_1 \to \mathcal{L}$ is a causal implicit generative model for the causal graph $D = (\mathcal{V}, E)$ where $\mathcal{V}$ is the set of image labels and the observational joint distribu-*

*tion over these labels are strictly positive. Let $G : \mathcal{L} \times \mathcal{Z}_2 \rightarrow \mathcal{I}$ be a generator that can sample from the image distribution conditioned on the given label combination $L \in \mathcal{L}$. Then $G(C(Z_1), Z_2)$ is a causal implicit generative model for the causal graph $D' = (\mathcal{V} \cup \{Image\}, E \cup \{(V_1, Image), (V_2, Image), \ldots (V_n, Image)\})$.*

*Proof.* Since $C$ is a causal implicit generative model for the causal graph $D$, by definition it is consistent with the causal graph $D$. Since in a conditional GAN, generator $G$ is given the noise terms and the labels, it is easy to see that the concatenated generator neural network $G(C(Z_1), Z_2)$ is consistent with the causal graph $D'$, where $D' = (\mathcal{V} \cup \{Image\}, E \cup \{(V_1, Image), (V_2, Image), \ldots (V_n, Image)\})$. Assume that $C$ and $G$ are perfect, i.e., they sample from the true label joint distribution and conditional image distribution. Then the joint distribution over the generated labels and image is the true distribution since $\mathbb{P}(Image, Label) = \mathbb{P}(Image|Label)\mathbb{P}(Label)$. By Proposition 1, the concatenated model can sample from the true observational and interventional distributions. Hence, the concatenated model is a causal implicit generative model for graph $D'$. □

## 8.6 CausalGAN Analysis for Multiple Labels

In this section, we explain the modifications required to extend the proof to the case with multiple binary labels. The central difficulty with generalizing to a vector of labels $l = (l_j)_{1 \le j \le d}$ is that each labeler can only hope to learn about the posterior $\mathbb{P}(l_j|x)$ *for each j*. This is in general insufficient to characterize $\mathbb{P}_r(l|x)$ and therefore the generator can not hope to learn the correct joint distribution. We show two solutions to this problem. (1) From a theoretical (but perhaps impractical) perspective each labeler can be made to estimate the probability of each of the $2^d$ label combinations instead of each label. We do not adopt this in practice. (2) If in fact the label vector is a deterministic function of the image (which seems likely for the present application), then using Labelers to estimate the probabilities of each of the $d$ labels is sufficient to assure $\mathbb{P}_g(l_1, l_2, \ldots, l_d, x) = \mathbb{P}_r(l_1, l_2, \ldots, l_d, x)$ at the minimizer of $C(G)$. In this section, we present the extension in (1) and present the results of (2) in Section 8.7.

Consider Figure 3 in the main text. The Labeler outputs the scalar $D_{LR}(x)$ given an image $x$. Previously in Section 8.3 we showed that the optimum Labeler satisfies $D_{LR}^*(x) = \mathbb{P}_r(l = 1|X = x)$ for a single label. We first extend the Labeler objective as follows: Suppose we have $d$ binary labels. Then we allow the Labeler to output a $2^d$ dimensional vector $D_{LR}(x)$, where $D_{LR}(x)[j]$ is the $j^{th}$ coordinate of this vector. The Labeler then solves the following optimization problem:

$$\max_{D_{LR}} \sum_{j=1}^{2^d} \rho_j \mathbb{E}_{x \sim \mathbb{P}_r(x|l=j)} \log(D_{LR}(x)[j]), \tag{12}$$

where $\rho_j = \mathbb{P}_r(l = j)$. We have the following Lemma:

**Lemma 3.** *Consider a Labeler $D_{LR}$ that outputs the $2^d$-dimensional vector $D_{LR}(x)$ such that $\sum_{j=1}^{2^d} D_{LR}(x)[j] = 1$, where $x \sim \mathbb{P}_r(x, l)$. Then the optimum Labeler with respect to the loss in (12) has $D_{LR}^*(x)[j] = \mathbb{P}_r(l = j|x)$.*

*Proof.* Suppose $\mathbb{P}_r(l = j|x) = 0$ for a set of (label, image) combinations. Then $\mathbb{P}_r(x, l = j) = 0$, hence these label combinations do not contribute to the expectation. Thus, without loss of generality, we can consider only the combinations with strictly positive probability. We can also restrict our attention to the functions $D_{LR}$ that are strictly positive on these (label,image) combinations; otherwise, loss becomes infinite, and as we will show we can achieve a finite loss. Consider the vector $D_{LR}(x)$ with coordinates $D_{LR}(x)[j]$ where $j \in [2^d]$. Introduce the discrete random variable $Z_x \in [2^d]$, where $\mathbb{P}(Z_x = j) = D_{LR}(x)[j]$. The Labeler loss can be written as

$$\min -\mathbb{E}_{(x,l) \sim \mathbb{P}_r(x,l)} \log(\mathbb{P}(Z_x = j)) \tag{13}$$

$$= \min \mathbb{E}_{x \sim \mathbb{P}_r(x)} KL(L_x \parallel Z_x) - H(L_x), \tag{14}$$

where $L_x$ is the discrete random variable such that $\mathbb{P}(L_x = j) = \mathbb{P}_r(l = j|x)$. $H(L_x)$ is the Shannon entropy of $L_x$, and it only depends on the data. Since KL divergence is greater than zero and $p(x)$ is always non-negative, the loss is lower bounded by $-H(L_x)$. Notice that this minimum

can be achieved by satisfying $\mathbb{P}(Z_x = j) = \mathbb{P}_r(l = j|x)$. Since KL divergence is minimized if and only if the two random variables have the same distribution, this is the unique optimum, i.e., $D^*_{LR}(x)[j] = \mathbb{P}_r(l = j|x)$.

$\square$

The lemma above simply states that the optimum Labeler network will give the posterior probability of a particular label combination, given the observed image. In practice, the constraint that the coordinates sum to 1 could be satisfied by using a softmax function in the implementation. Next, we have the corresponding loss function and lemma for the Anti-Labeler network. The Anti-Labeler solves the following optimization problem

$$\max_{D_{LG}} \sum_{j=1}^{2^d} \rho_j \mathbb{E}_{\mathbb{P}_g(x|l=j)} \log(D_{LG}(x)[j]), \tag{15}$$

where $\mathbb{P}_g(x|l = j) := \mathbb{P}(G(z, l) = x|l = j)$ and $\rho_j = \mathbb{P}(l = j)$. We have the following Lemma:

**Lemma 4.** *The optimum Anti-Labeler has* $D^*_{LG}(x)[j] = \mathbb{P}_g(l = j|x)$.

*Proof.* The proof is the same as the proof of Lemma 3, since Anti-Labeler does not have control over the joint distribution between the generated image and the labels given to the generator, and cannot optimize the conditional entropy of labels given the image under this distribution. $\square$

For a fixed discriminator, Labeler and Anti-Labeler, the generator solves the following optimization problem:

$$\min_G \mathbb{E}_{x \sim p_g(x)} \left[ \log \left( \frac{1 - D(x)}{D(x)} \right) \right]$$

$$- \sum_{j=1}^{2^d} \rho_j \mathbb{E}_{x \sim \mathbb{P}_g(x|l=j)} \left[ \log(D_{LR}(X)[j]) \right]$$

$$+ \sum_{j=1}^{2^d} \rho_j \mathbb{E}_{x \sim \mathbb{P}_g(x|l=j)} \left[ \log(D_{LG}(X)[j]) \right]. \tag{16}$$

We then have the following theorem along the same lines as Theorem 1 showing that the optimal generator samples from the class conditional image distributions given a particular label combination:

**Theorem 2** (Theorem 1 formal for multiple binary labels)**.** *Define $C(G)$ as the generator loss as in Eqn. 16 when discriminator, Labeler and Anti-Labeler are at their optimum. Assume $\mathbb{P}_g(l) = \mathbb{P}_r(l)$, i.e., the Causal Controller samples from the true joint label distribution. The global minimum of the virtual training criterion $C(G)$ is achieved if and only if $\mathbb{P}_g(l, x) = \mathbb{P}_r(l, x)$ for the vector of labels $l = \{l_i\}_{1 \leq i \leq 2^d}$.*

*Proof.* For a fixed generator, the optimum Labeler $D^*_{LR}$, Anti-Labeler $D^*_{LG}$, and discriminator $D^*$ obey the following relations by Prop 2, Lemma 3, and Lemma 4:

$$(1 - D^*(x))/D^*(x) = \mathbb{P}_g(x)/\mathbb{P}_r(x)$$
$$D^*_{LR}(x)[j] = \mathbb{P}_r(l = j|x) \; \forall j$$
$$D^*_{LG}(x)[j] = \mathbb{P}_g(l = 1|x) \; \forall j.$$

$$\tag{17}$$

Then substitution into the generator objective $C(G)$ yields

$$C(G) = \sum_{j=1}^{2^d} \rho_j \mathbb{E}_{x \sim \mathbb{P}_g(x|l=j)} \left[ \log \left( \frac{\mathbb{P}_g(x)}{\mathbb{P}_r(x)} \right) + \log(\mathbb{P}_g(l=j|x)) - \log(\mathbb{P}_r(l=j|x)) \right]$$

$$= \sum_{j=1}^{2^d} \rho_j \mathbb{E}_{x \sim \mathbb{P}_g(x|l=j)} \left[ \log \left( \frac{\mathbb{P}_g(l=j,x)}{\mathbb{P}_r(l=j,x)} \right) \right]$$

$$= \mathbb{E}_{(l,x) \sim \mathbb{P}_g(l,x)} \left[ \log \left( \frac{\mathbb{P}_g(l,x)}{\mathbb{P}_d(l,x)} \right) \right]$$

$$= KL(\mathbb{P}_g \parallel \mathbb{P}_d). \tag{18}$$

where KL is the Kullback-Leibler divergence, which is minimized if and only if $\mathbb{P}_g = \mathbb{P}_d$ jointly over labels and images.

$\square$

## 8.7 CAUSALGAN EXTENSION TO $d$ LABELS UNDER DETERMINISTIC LABELS

While the previous section showed how to ensure $\mathbb{P}_g(l, x) = \mathbb{P}_r(l, x)$ by relabeling combinations of a $d$ binary labels as a $2^d$ label, this may be difficult in practice for a large number of labels and we do not adopt this approach in practice.

Instead, in this section, we provide the theoretical guarantees for the implemented CausalGAN architecture with $d$ labels under the assumption that the relationship between the image and its labels is deterministic in the dataset, i.e., there is a deterministic function that maps an image to the corresponding label vector. Later we show that this assumption is sufficient to gaurantee that the global optimal generator samples from the class conditional distributions.

First, let us restate the loss functions more formally. Note that $D_{LR}(x), D_{LG}(x)$ are $d-$dimensional vectors. The Labeler solves the following optimization problem:

$$\max_{D_{LR}} \rho_j \mathbb{E}_{x \sim \mathbb{P}_r(x|l_j=1)} \log(D_{LR}(x)[j]) + (1 - \rho_j) \mathbb{E}_{x \sim \mathbb{P}_r(x|l_j=0)} \log(1 - D_{LR}(x)[j]). \tag{19}$$

where $\mathbb{P}_r(x|l_j = 0) := \mathbb{P}(X = x|l_j = 0)$, $\mathbb{P}_r(x|l_j = 0) := \mathbb{P}(X = x|l_j = 0)$ and $\rho_j = \mathbb{P}(l_j = 1)$. For a fixed generator, the Anti-Labeler solves the following optimization problem:

$$\max_{D_{LG}} \rho_j \mathbb{E}_{\mathbb{P}_g(x|l_j=1)} \log(D_{LG}(x)[j]) + (1 - \rho_j) \mathbb{E}_{\mathbb{P}_g(x|l_j=0)} \log(1 - D_{LG}(x)[j]), \tag{20}$$

where $\mathbb{P}_g(x|l_j = 0) := \mathbb{P}_g(x|l_j = 0)$, $\mathbb{P}_g(x|l_j = 0) := \mathbb{P}_g(x|l_j = 0)$. For a fixed discriminator, Labeler and Anti-Labeler, the generator solves the following optimization problem:

$$\min_G \mathbb{E}_{x \sim p_{\text{data}}(x)} \left[ \log(D(x)) \right] + \mathbb{E}_{x \sim p_g(x)} \left[ \log \left( \frac{1 - D(x)}{D(x)} \right) \right]$$

$$- \frac{1}{d} \sum_{j=1}^d \rho_j \mathbb{E}_{x \sim \mathbb{P}_g(x|l_j=1)} \left[ \log(D_{LR}(X)[j]) \right] - (1 - \rho_j) \mathbb{E}_{x \sim \mathbb{P}_g(x|l_j=0)} \left[ \log(1 - D_{LR}(X)[j]) \right]$$

$$+ \frac{1}{d} \sum_{j=1}^d \rho_j \mathbb{E}_{x \sim \mathbb{P}_g(x|l_j=1)} \left[ \log(D_{LG}(X)[j]) \right] + (1 - \rho_j) \mathbb{E}_{x \sim \mathbb{P}_g(x|l_j=0)} \left[ \log(1 - D_{LG}(X)[j]) \right]. \tag{21}$$

We have the following proposition, which characterizes the optimum generator for optimum Labeler, Anti-Labeler and Discriminator:

**Proposition 3.** *Define $C(G)$ as the generator loss for when discriminator, Labeler and Anti-Labeler are at their optimum obtained from (21). The global minimum of the virtual training criterion $C(G)$ is achieved if and only if $\mathbb{P}_g(x|l_i) = \mathbb{P}_r(x|l_i) \forall i \in [d]$ and $\mathbb{P}_g(x) = \mathbb{P}_r(x)$.*

*Proof.* Proof follows the same lines as in the proof of Theorem 1 and Theorem 2 and is omitted. □

Thus we have
$$\mathbb{P}_r(x, l_i) = \mathbb{P}_g(x, l_i), \forall i \in [d] \text{ and } \mathbb{P}_r(x) = \mathbb{P}_g(x). \tag{22}$$
However, this does not in general imply $\mathbb{P}_r(x, l_1, l_2, \ldots, l_d) = \mathbb{P}_g(x, l_1, l_2, \ldots, l_d)$, which is equivalent to saying the generated distribution samples from the class conditional image distributions. To guarantee the correct conditional sampling given all labels, we introduce the following assumption: We assume that the image $x$ determines all the labels. This assumption is very relevant in practice. For example, in the CelebA dataset, which we use, the label vector, e.g., whether the person is a male or female, with or without a mustache, can be thought of as a deterministic function of the image. When this is true, we can say that $\mathbb{P}_r(l_1, l_2, \ldots, l_n|x) = \mathbb{P}_r(l_1|x)\mathbb{P}_r(l_2|x)\ldots\mathbb{P}_r(l_n|x)$.

We need the following lemma, where kronecker delta function refers to the functions that take the value of 1 only on a single point, and 0 everywhere else:

**Lemma 5.** *Any discrete joint probability distribution, where all the marginal probability distributions are kronecker delta functions is the product of these marginals.*

*Proof.* Let $\delta_{\{x-u\}}$ be the kronecker delta function which is 1 if $x = u$ and is 0 otherwise. Consider a joint distribution $p(X_1, X_2, \ldots, X_n)$, where $p(X_i) = \delta_{\{X_i - u_i\}}, \forall i \in [n]$, for some set of elements $\{u_i\}_{i \in [n]}$. We will show by contradiction that the joint probability distribution is zero everywhere except at $(u_1, u_2, \ldots, u_n)$. Then, for the sake of contradiction, suppose for some $v = (v_1, v_2, \ldots, v_n) \neq (u_1, u_2, \ldots, u_n)$, $p(v_1, v_2, \ldots, v_n) \neq 0$. Then $\exists j \in [n]$ such that $v_j \neq u_j$. Then we can marginalize the joint distribution as

$$p(v_j) = \sum_{X_1, \ldots, X_{j-1}, X_j, \ldots, X_n} p(X_1, \ldots, X_{j-1}, v_j, X_{j+1}, \ldots, X_n) > 0, \tag{23}$$

where the inequality is due to the fact that the particular configuration $(v_1, v_2, \ldots, v_n)$ must have contributed to the summation. However this contradicts with the fact that $p(X_j) = 0, \forall X_j \neq u_j$. Hence, $p(.)$ is zero everywhere except at $(u_1, u_2, \ldots, u_n)$, where it should be 1. □

We can now simply apply the above lemma on the conditional distribution $\mathbb{P}_g(l_1, l_2, \ldots, l_d|x)$. Proposition 3 shows that the image distributions and the marginals $\mathbb{P}_g(l_i|x)$ are true to the data distribution due to Bayes' rule. Since the vector $(l_1, \ldots, l_n)$ is a deterministic function of $x$ by assumption, $\mathbb{P}_r(l_i|x)$ are kronecker delta functions, and so are $\mathbb{P}_g(l_i|x)$ by Proposition 3. Thus, since the joint $\mathbb{P}_g(x, l_1, l_2, \ldots, l_d)$ satisfies the condition that every marginal distribution $p(l_i|x)$ is a kronecker delta function, then it must be a product distribution by Lemma 5. Thus we can write

$$\mathbb{P}_g(l_1, l_2, \ldots, l_d|x) = \mathbb{P}_g(l_1|x)\mathbb{P}_g(l_2|x)\ldots\mathbb{P}_g(l_n|x).$$

Then we have the following chain of equalities.

$$\begin{aligned}
\mathbb{P}_r(x, l_1, l_2, \ldots, l_d) &= \mathbb{P}_r(l_1, \ldots, l_n|x)\mathbb{P}_r(x) \\
&= \mathbb{P}_r(l_1|x)\mathbb{P}_r(l_2|x)\ldots\mathbb{P}_r(l_n|x)\mathbb{P}_r(x) \\
&= \mathbb{P}_g(l_1|x)\mathbb{P}_g(l_2|x)\ldots\mathbb{P}_g(l_n|x)\mathbb{P}_g(x) \\
&= \mathbb{P}_g(l_1, l_2, \ldots, l_d|x)\mathbb{P}_g(x) \\
&= \mathbb{P}_g(x, l_1, l_2, \ldots, l_d).
\end{aligned}$$

Thus, we also have $\mathbb{P}_r(x|l_1, l_2, \ldots, l_n) = \mathbb{P}_g(x|l_1, l_2, \ldots, l_n)$ since $\mathbb{P}_r(l_1, l_2, \ldots, l_n) = \mathbb{P}_g(l_1, l_2, \ldots, l_n)$, concluding the proof that the optimum generator samples from the class conditional image distributions.

## 8.8 CAUSALBEGAN ARCHITECTURE

In this section, we propose a simple, but non-trivial extension of BEGAN where we feed image labels to the generator. One of the central contributions of BEGAN (Berthelot et al. (2017)) is a control theory-inspired boundary equilibrium approach that encourages generator training only when the discriminator is near optimum and its gradients are the most informative. The following observation

helps us carry the same idea to the case with labels: Label gradients are most informative when the image quality is high. Here, we introduce a new loss and a set of margins that reflect this intuition.

Formally, let $\mathcal{L}(x)$ be the average $L_1$ pixel-wise autoencoder loss for an image $x$, as in BEGAN. Let $\mathcal{L}_{sq}(u, v)$ be the squared loss term, i.e., $\|u - v\|_2^2$. Let $(x, l_x)$ be a sample from the data distribution, where $x$ is the image and $l_x$ is its corresponding label. Similarly, $G(z, l_g)$ is an image sample from the generator, where $l_g$ is the label used to generate this image. Denoting the space of images by $\mathcal{I}$, let $G : \mathbb{R}^n \times \{0, 1\}^m \mapsto \mathcal{I}$ be the generator. As a naive attempt to extend the original BEGAN loss formulation to include the labels, we can write the following loss functions:

$$Loss_D = \mathcal{L}(x) - \mathcal{L}(Labeler(G(z, l))) + \mathcal{L}_{sq}(l_x, Labeler(x)) - \mathcal{L}_{sq}(l_g, Labeler(G(z, l_g))),$$
$$Loss_G = \mathcal{L}(G(z, l_g)) + \mathcal{L}_{sq}(l_g, Labeler(G(z, l_g))). \tag{24}$$

However, this naive formulation does not address the use of margins, which is extremely critical in the BEGAN formulation. Just as a better trained BEGAN discriminator creates more useful gradients for image generation, a better trained Labeler is a prerequisite for meaningful gradients. This motivates an additional margin-coefficient tuple $(b_2, c_2)$, as shown in (25,26).

The generator tries to jointly minimize the two loss terms in the formulation in (24). We empirically observe that occasionally the image quality will suffer because the images that best exploit the Labeler network are often not obliged to be realistic, and can be noisy or misshapen. Based on this, label loss seems unlikely to provide useful gradients unless the image quality remains good. Therefore we encourage the generator to incorporate label loss only when the *image quality margin $b_1$* is large compared to the *label margin $b_2$*. To achieve this, we introduce a new *margin of margins* term, $b_3$. As a result, the margin equations and update rules are summarized as follows, where $\lambda_1, \lambda_2, \lambda_3$ are learning rates for the coefficients.

$$b_1 = \gamma_1 * \mathcal{L}(x) - \mathcal{L}(G(z, l_g)).$$
$$b_2 = \gamma_2 * \mathcal{L}_{sq}(l_x, Labeler(x)) - \mathcal{L}_{sq}(l_g, Labeler(G(z, l_g))). \tag{25}$$
$$b_3 = \gamma_3 * relu(b_1) - relu(b_2).$$
$$c_1 \leftarrow clip_{[0,1]}(c_1 + \lambda_1 * b_1).$$
$$c_2 \leftarrow clip_{[0,1]}(c_2 + \lambda_2 * b_2). \tag{26}$$
$$c_3 \leftarrow clip_{[0,1]}(c_3 + \lambda_3 * b_3).$$
$$Loss_D = \mathcal{L}(x) - c_1 * \mathcal{L}(G(z, l_g)) + \mathcal{L}_{sq}(l_x, Labeler(x)) - c_2 * \mathcal{L}_{sq}(l_g, G(z, l_g)). \tag{27}$$
$$Loss_G = \mathcal{L}(G(z, l_g)) + c_3 * \mathcal{L}_{sq}(l_g, Labeler(G(z, l_g))).$$

One of the advantages of BEGAN is the existence of a monotonically decreasing scalar which can track the convergence of the gradient descent optimization. Our extension preserves this property as we can define

$$\mathcal{M}_{complete} = \mathcal{L}(x) + |b_1| + |b_2| + |b_3|, \tag{28}$$

and show that $\mathcal{M}_{complete}$ decreases progressively during our optimizations. See Figure 19.

## 8.9 Dependence of GAN Behavior on Causal Graph

In Section 4 we showed how a GAN could be used to train a causal implicit generative model by incorporating the causal graph into the generator structure. Here we investigate the behavior and convergence of causal implicit generative models when the true data distribution arises from another (possibly distinct) causal graph.

We consider causal implicit generative model convergence on synthetic data whose three features $\{X, Y, Z\}$ arise from one of three causal graphs: "line" $X \to Y \to Z$, "collider" $X \to Y \leftarrow Z$, and "complete" $X \to Y \to Z, X \to Z$. For each node a (randomly sampled once) cubic polynomial in $n + 1$ variables computes the value of that node given its $n$ parents and 1 uniform exogenous variable. We then repeat, creating a new synthetic dataset in this way for each causal model and report the averaged results of 20 runs for each model.

For each of these data generating graphs, we compare the convergence of the joint distribution to the true joint in terms of the total variation distance, when the generator is structured according to a

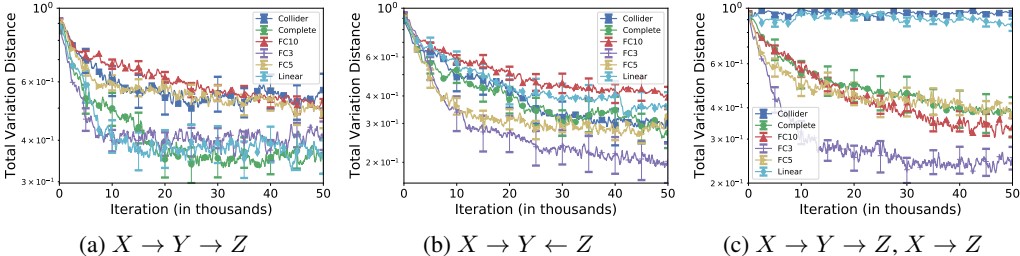

(a) $X \to Y \to Z$          (b) $X \to Y \leftarrow Z$          (c) $X \to Y \to Z, X \to Z$

Figure 9: Convergence in total variation distance of generated distribution to the true distribution for causal implicit generative model, when the generator is structured based on different causal graphs. (a) Data generated from line graph $X \to Y \to Z$. The best convergence behavior is observed when the true causal graph is used in the generator architecture. (b) Data generated from collider graph $X \to Y \leftarrow Z$. Fully connected layers may perform better than the true graph depending on the number of layers. Collider and complete graphs performs better than the line graph which implies the wrong Bayesian network. (c) Data generated from complete graph $X \to Y \to Z, X \to Z$. Fully connected with 3 layers performs the best, followed by the complete and fully connected with 5 and 10 layers. Line and collider graphs, which implies the wrong Bayesian network does not show convergence behavior.

line, collider, or complete graph. For completeness, we also include generators with no knowledge of causal structure: $\{fc3, fc5, fc10\}$ are fully connected neural networks that map uniform random noise to 3 output variables using either 3,5, or 10 layers respectively.

The results are given in Figure 9. Data is generated from line causal graph $X \to Y \to Z$ (left panel), collider causal graph $X \to Y \leftarrow$ (middle panel), and complete causal graph $X \to Y \to Z, X \to Z$ (right panel). Each curve shows the convergence behavior of the generator distribution, when generator is structured based on each one of these causal graphs. We expect convergence when the causal graph used to structure the generator is capable of generating the joint distribution due to the true causal graph: as long as we use the correct Bayesian network, we should be able to fit to the true joint. For example, complete graph can encode all joint distributions. Hence, we expect complete graph to work well with all data generation models. Standard fully connected layers correspond to the causal graph with a latent variable causing all the observable variables. Ideally, this model should be able to fit to any causal generative model. However, the convergence behavior of adversarial training across these models is unclear, which is what we are exploring with Figure 9.

For the line graph data $X \to Y \to Z$, we see that the best convergence behavior is when line graph is used in the generator architecture. As expected, complete graph also converges well, with slight delay. Similarly, fully connected network with 3 layers show good performance, although surprisingly fully connected with 5 and 10 layers perform much worse. It seems that although fully connected can encode the joint distribution in theory, in practice with adversarial training, the number of layers should be tuned to achieve the same performance as using the true causal graph. Using the wrong Bayesian network, the collider, also yields worse performance.

For the collider graph, surprisingly using a fully connected generator with 3 and 5 layers shows the best performance. However, consistent with the previous observation, the number of layers is important, and using 10 layers gives the worst convergence behavior. Using complete and collider graphs achieves the same decent performance, whereas line graph, a wrong Bayesian network, performs worse than the two.

For the complete graph, fully connected 3 performs the best, followed by fully connected 5, 10 and the complete graph. As we expect, line and collider graphs, which cannot encode all the distributions due to a complete graph, performs the worst and does not actually show any convergence behavior.

## 8.10 ADDITIONAL SIMULATIONS FOR CAUSAL CONTROLLER

First, we evaluate the effect of using the wrong causal graph on an artificially generated dataset. Figure 10 shows the scatter plot for the two coordinates of a three dimensional distribution. As we

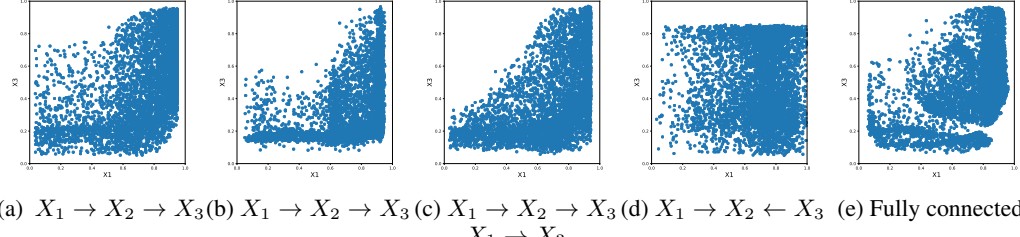

(a) $X_1 \to X_2 \to X_3$ (b) $X_1 \to X_2 \to X_3$ (c) $X_1 \to X_2 \to X_3$ (d) $X_1 \to X_2 \leftarrow X_3$ (e) Fully connected
$X_1 \to X_3$

Figure 10: Synthetic data experiments: (a) Scatter plot for actual data. Data is generated using the causal graph $X_1 \to X_2 \to X_3$. (b) Generated distribution when generator causal graph is $X_1 \to X_2 \to X_3$. (c) Generated distribution when generator causal graph is $X_1 \to X_2 \to X_3 \cup X_1 \to X_3$. (d) Generated distribution when generator causal graph is $X_1 \to X_2 \leftarrow X_3$. (e) Generated distribution when generator is from a fully connected last layer of a 5 layer FF neural net.

| Label | Male | | |
|---|---|---|---|
| Pair | | 0 | 1 |
| Young | 0 | 0.14[0.07](0.07) | 0.09[0.15](0.15) |
| | 1 | 0.47[0.51](0.51) | 0.29[0.27](0.26) |
| Mustache | 0 | 0.61[0.58](0.58) | 0.34[0.38](0.38) |
| | 1 | 0.00[0.00](0.00) | 0.04[0.04](0.04) |

Table 1: Pairwise marginal distribution for select label pairs when Causal Controller is trained on $G1$ in plain text, its completion $cG1$[square brackets], and the true pairwise distribution(in parentheses). Note that $G1$ treats Male and Young labels as independent, but does not completely fail to generate a reasonable (product of marginals) approximation. Also note that when an edge is added $Young \to Male$, the learned distribution is nearly exact. Note that both graphs contain the edge $Male \to Mustache$ and so are able to learn that women have no mustaches.

observe, using the correct graph gives the closest scatter plot to the original data, whereas using the wrong Bayesian network, collider graph, results in a very different distribution.

Second, we expand on the causal graphs used for experiments for the CelebA dataset. We use a causal graph on a subset of the image labels of CelebA dataset, which we call CelebA Causal Graph (G1), illustrated in Figure 8. The graph cG1, which is a completed version of G1, is the complete graph associated with the ordering: Young, Male, Eyeglasses, Bald, Mustache, Smiling, Wearing Lipstick, Mouth Slightly Open, Narrow Eyes. For example, in cG1 Male causes Smiling because Male comes before Smiling in the ordering. The graph rcG1 is formed by reversing every edge in cG1.

Next, we check the effect of using the incorrect Bayesian network for the data. The causal graph G1 generates Male and Young independently, which is incorrect in the data. Comparison of pairwise distributions in Table 1 demonstrate that for G1 a reasonable approximation to the true distribution is still learned for {Male, Young} jointly. For cG1 a nearly perfect distributional approximation is learned. Furthermore we show that despite this inaccuracy, both graphs G1 and cG1 lead to Causal Controllers that never output the label combination {Female,Mustache}, which will be important later.

Wasserstein GAN in its original form (with Lipshitz discriminator) assures convergence in distribution of the Causal Controller output to the discretely supported distribution of labels. We use a slightly modified version of Wasserstein GAN with a penalized gradient (Gulrajani et al. (2017)). We first demonstrate that learned outputs actually have "approximately discrete" support. In Figure 11a, we sample the joint label distribution 1000 times, and make a histogram of the (all) scalar outputs corresponding to any label.

Although Figure 11b demonstrates conclusively good convergence for both graphs, TVD is not always intuitive. For example, "how much can each marginal be off if there are 9 labels and the TVD

| Label, L | $\mathbb{P}_{G1}(L=1)$ | $\mathbb{P}_{cG1}(L=1)$ | $\mathbb{P}_D(L=1)$ |
|---|---|---|---|
| Bald | 0.02244 | 0.02328 | 0.02244 |
| Eyeglasses | 0.06180 | 0.05801 | 0.06406 |
| Male | 0.38446 | 0.41938 | 0.41675 |
| Mouth Slightly Open | 0.49476 | 0.49413 | 0.48343 |
| Mustache | 0.04596 | 0.04231 | 0.04154 |
| Narrow Eyes | 0.12329 | 0.11458 | 0.11515 |
| Smiling | 0.48766 | 0.48730 | 0.48208 |
| Wearing Lipstick | 0.48111 | 0.46789 | 0.47243 |
| Young | 0.76737 | 0.77663 | 0.77362 |

Table 2: Marginal distribution of pretrained Causal Controller labels when Causal Controller is trained on CelebA Causal Graph ($P_{G1}$) and its completion($P_{cG1}$), where $cG1$ is the (nonunique) largest DAG containing $G1$ (see appendix). The third column lists the actual marginal distributions in the dataset

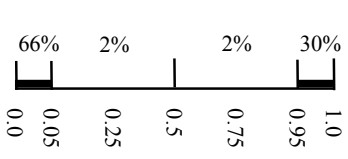

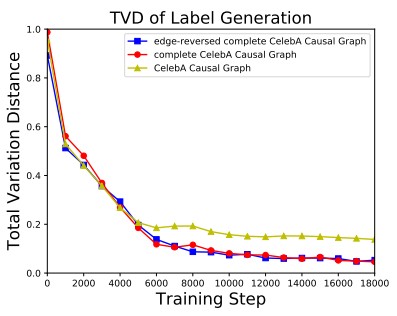

(a) Essentially Discrete Range of Causal Controller  (b) TVD vs. No. of Iters in CelebA Labels

Figure 11: (a) A number line of unit length binned into 4 unequal bins along with the percent of Causal Controller ($G1$) samples in each bin. Results are obtained by sampling the joint label distribution 1000 times and forming a histogram of the scalar outputs corresponding to any label. Note that our Causal Controller output labels are approximately discrete even though the input is a continuum (uniform). The 4% between 0.05 and 0.95 is not at all uniform and almost zero near 0.5. (b) Progression of total variation distance between the Causal Controller output with respect to the number of iterations: CelebA Causal Graph is used in the training with Wasserstein loss.

is 0.14?". To expand upon Figure 2 where we showed that the causal controller learns the correct distribution for a pairwise subset of nodes, here we also show that both CelebA Causal Graph (G1) and the completion we define (cG1) allow training of very reasonable marginal distributions for all labels (Table 1) that are not off by more than 0.03 for the worst label. $\mathbb{P}_D(L=1)$ is the probability that the label is 1 in the dataset, and $\mathbb{P}_G(L=1)$ is the probability that the generated label is (around a small neighborhood of ) 1.

## 8.11 WASSERSTEIN CAUSAL CONTROLLER ON CELEBA LABELS

We test the performance of our Wasserstein Causal Controller on a subset of the binary labels of CelebA datset. We use the causal graph given in Figure 8.

For causal graph training, first we verify that our Wasserstein training allows the generator to learn a mapping from continuous uniform noise to a discrete distribution. Figure 11a shows where the samples, averaged over all the labels in CelebA Causal Graph, from this generator appears on the real line. The result emphasizes that the proposed Causal Controller outputs an almost discrete distribution: 96% of the samples appear in $0.05-$neighborhood of 0 or 1. Outputs shown are *unrounded* generator outputs.

A stronger measure of convergence is the total variational distance (TVD). For CelebA Causal Graph (G1), our defined completion (cG1), and cG1 with arrows reversed (rcG1), we show convergence of TVD with training (Figure 11b). Both cG1 and rcG1 have TVD decreasing to 0, and TVD for G1 assymptotes to around 0.14 which corresponds to the incorrect conditional independence assumptions that G1 makes. This suggests that any given complete causal graph will lead to a nearly perfect implicit causal generator over labels and that bayesian partially incorrect causal graphs can still give reasonable convergence.

## 8.12 MORE CAUSALGAN RESULTS

In this section, we present additional CausalGAN results in Figure 12, 13.

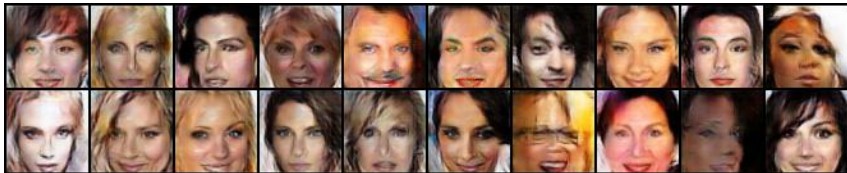

Intervening vs Conditioning on Wearing Lipstick, Top: Intervene Wearing Lipstick=1, Bottom: Condition Wearing Lipstick=1

Figure 12: Intervening/Conditioning on Wearing Lipstick label in CelebA Causal Graph. Since $Male \rightarrow WearingLipstick$ in CelebA Causal Graph, we do not expect $do(Wearing\ Lipstick = 1)$ to affect the probability of $Male = 1$, i.e., $\mathbb{P}(Male = 1|do(Wearing\ Lipstick = 1)) = \mathbb{P}(Male = 1) = 0.42$. Accordingly, the top row shows both males and females who are wearing lipstick. However, the bottom row of images sampled from the conditional distribution $\mathbb{P}(.|Wearing\ Lipstick = 1)$ shows only female images because in the dataset $\mathbb{P}(Male = 0|Wearing\ Lipstick = 1) \approx 1$.

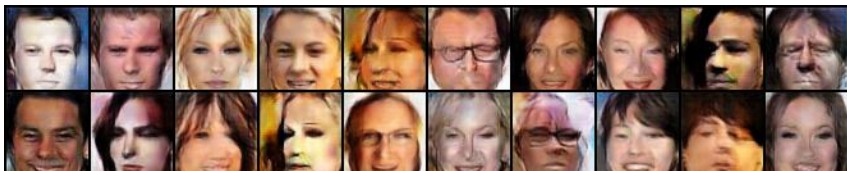

Intervening vs Conditioning on Narrow Eyes, Top: Intervene Narrow Eyes=1, Bottom: Condition Narrow Eyes=1

Figure 13: Intervening/Conditioning on Narrow Eyes label in CelebA Causal Graph. Since $Smiling \rightarrow Narrow\ Eyes$ in CelebA Causal Graph, we do not expect $do(Narrow\ Eyes = 1)$ to affect the probability of $Smiling = 1$, i.e., $\mathbb{P}(Smiling = 1|do(Narrow\ Eyes = 1)) = \mathbb{P}(Smiling = 1) = 0.48$. However on the bottom row, conditioning on *Narrow Eyes = 1* increases the proportion of smiling images (From $0.48$ to $0.59$ in the dataset), although 10 images may not be enough to show this difference statistically.

## 8.13 MORE CAUSALBEGAN RESULTS

In this section, we train CausalBEGAN on CelebA dataset using CelebA Causal Graph. The Causal Controller is pretrained with a Wasserstein loss and used for training the CausalBEGAN.

To first empirically justify the need for the margin of margins we introduced in (27) ($c_3$ and $b_3$), we train the same CausalBEGAN model setting $c_3 = 1$, removing the effect of this margin. We show that the image quality for rare labels deteriorates. Please see Figure 18 in the appendix. Then for the labels *Bald*, and *Mouth Slightly Open*, we illustrate the difference between interventional and conditional sampling when the label is 1. (Figures 14, 15).

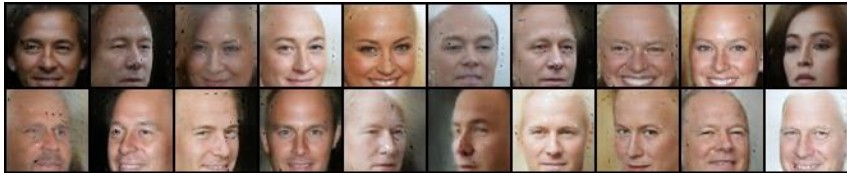

Intervening vs Conditioning on Bald, Top: Intervene Bald=1, Bottom: Condition Bald=1

Figure 14: Intervening/Conditioning on Bald label in CelebA Causal Graph. Since $Male \rightarrow Bald$ in CelebA Causal Graph, we do not expect $do(Bald = 1)$ to affect the probability of $Male = 1$, i.e., $\mathbb{P}(Male = 1|do(Bald = 1)) = \mathbb{P}(Male = 1) = 0.42$. Accordingly, the top row shows both bald males and bald females. The bottom row of images sampled from the conditional distribution $\mathbb{P}(.|Bald = 1)$ shows only male images because in the dataset $\mathbb{P}(Male = 1|Bald = 1) \approx 1$.

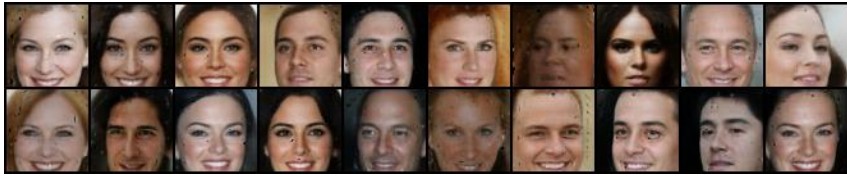

Intervening vs Conditioning on Mouth Slightly Open, Top: Intervene Mouth Slightly Open=1, Bottom: Condition Mouth Slightly Open=1

Figure 15: Intervening/Conditioning on Mouth Slightly Open label in CelebA Causal Graph. Since $Smiling \rightarrow MouthSlightlyOpen$ in CelebA Causal Graph, we do not expect $do(Mouth\ Slightly\ Open = 1)$ to affect the probability of $Smiling = 1$, i.e., $\mathbb{P}(Smiling = 1|do(Mouth\ Slightly\ Open = 1)) = \mathbb{P}(Smiling = 1) = 0.48$. However on the bottom row, conditioning on *Mouth Slightly Open* $= 1$ increases the proportion of smiling images (From $0.48$ to $0.76$ in the dataset), although 10 images may not be enough to show this difference statistically.

### 8.14 LABEL SWEEPING AND DIVERSITY FOR CAUSALGAN

In this section, we provide additional simulations for CausalGAN. In Figures 16a-16d, we show the conditional image generation properties of CausalGAN by sweeping a single label from 0 to 1 while keeping all other inputs/labels fixed. In Figure 17, to examine the degree of mode collapse and show the image diversity, we show 256 randomly sampled images.

### 8.15 ADDITIONAL CAUSALBEGAN SIMULATIONS

In this section, we provide additional simulation results for CausalBEGAN. First we show that although our third margin term $b_3$ introduces complications, it can not be ignored. Figure 18 demonstrates that omitting the third margin on the image quality of rare labels.

Furthermore just as the setup in BEGAN permitted the definiton of a scalar "$\mathcal{M}$", which was monotonically decreasing during training, our definition permits an obvious extension $\mathcal{M}_{complete}$ (defined in 28) that preserves these properties. See Figure 19 to observe $\mathcal{M}_{complete}$ decreaing monotonically during training.

We also show the conditional image generation properties of CausalBEGAN by using "label sweeps" that move a single label input from 0 to 1 while keeping all other inputs fixed (Figures 20a -20d ). It is interesting to note that while generators are often implicitly thought of as continuous functions, the generator in this CausalBEGAN architecture learns a discrete function with respect to its label input parameters. (Initially there is label interpolation, and later in the optimization label interpolation becomes more step function like (not shown)). Finally, to examine the degree of mode collapse and show the image diversity, we show a random sampling of 256 images (Figure 21).

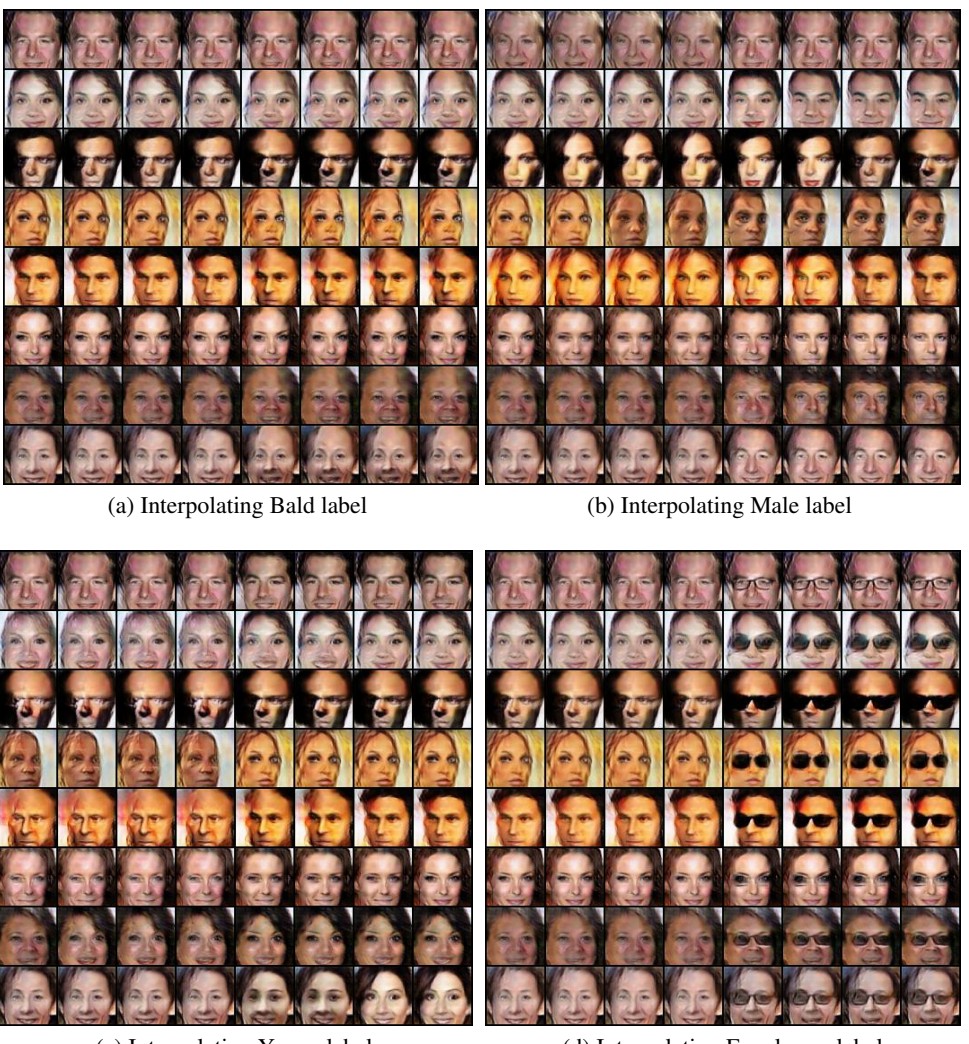

(a) Interpolating Bald label

(b) Interpolating Male label

(c) Interpolating Young label

(d) Interpolating Eyeglasses label

Figure 16: The effect of interpolating a single label for CausalGAN, while keeping the noise terms and other labels fixed.

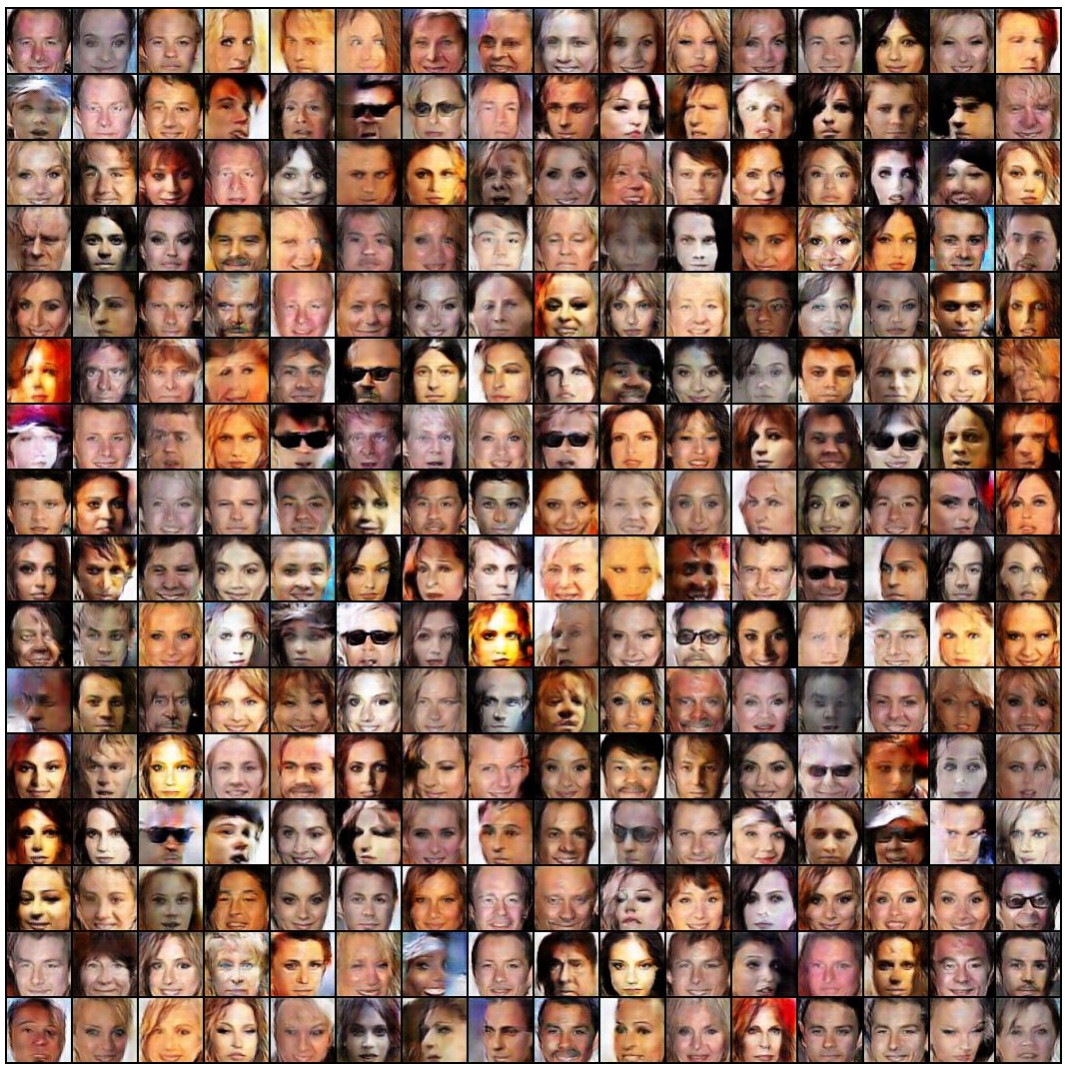

Figure 17: Diversity of the proposed CausalGAN showcased with 256 samples.

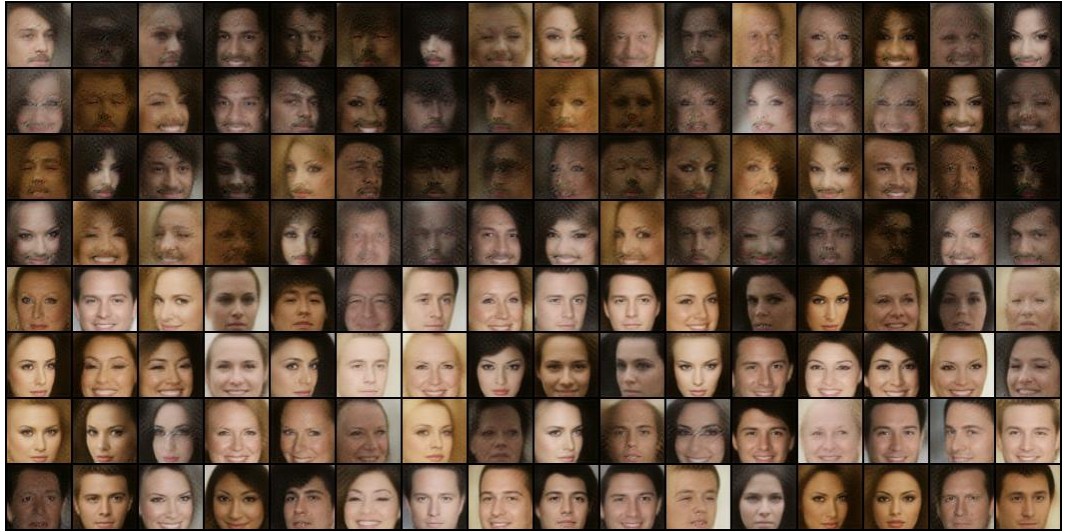

Figure 18: Omitting the nonobvious margin $b_3 = \gamma_3 * relu(b_1) - relu(b_2)$ results in poorer image quality particularly for rare labels such as mustache. We compare samples from two interventional distributions. Samples from $\mathbb{P}(.|do(Mustache = 1))$ (top) have much poorer image quality compared to those under $\mathbb{P}(.|do(Mustache = 0))$ (bottom).

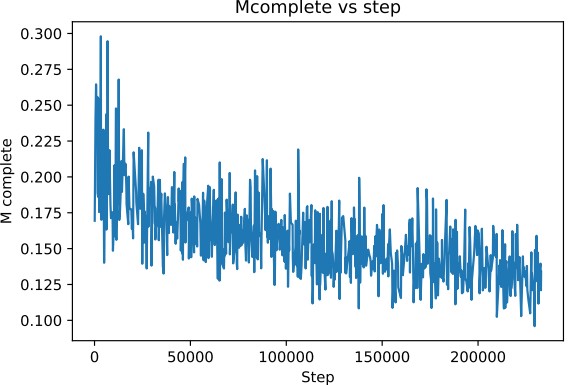

Figure 19: Convergence of CausalBEGAN captured through the parameter $\mathcal{M}_{complete}$.

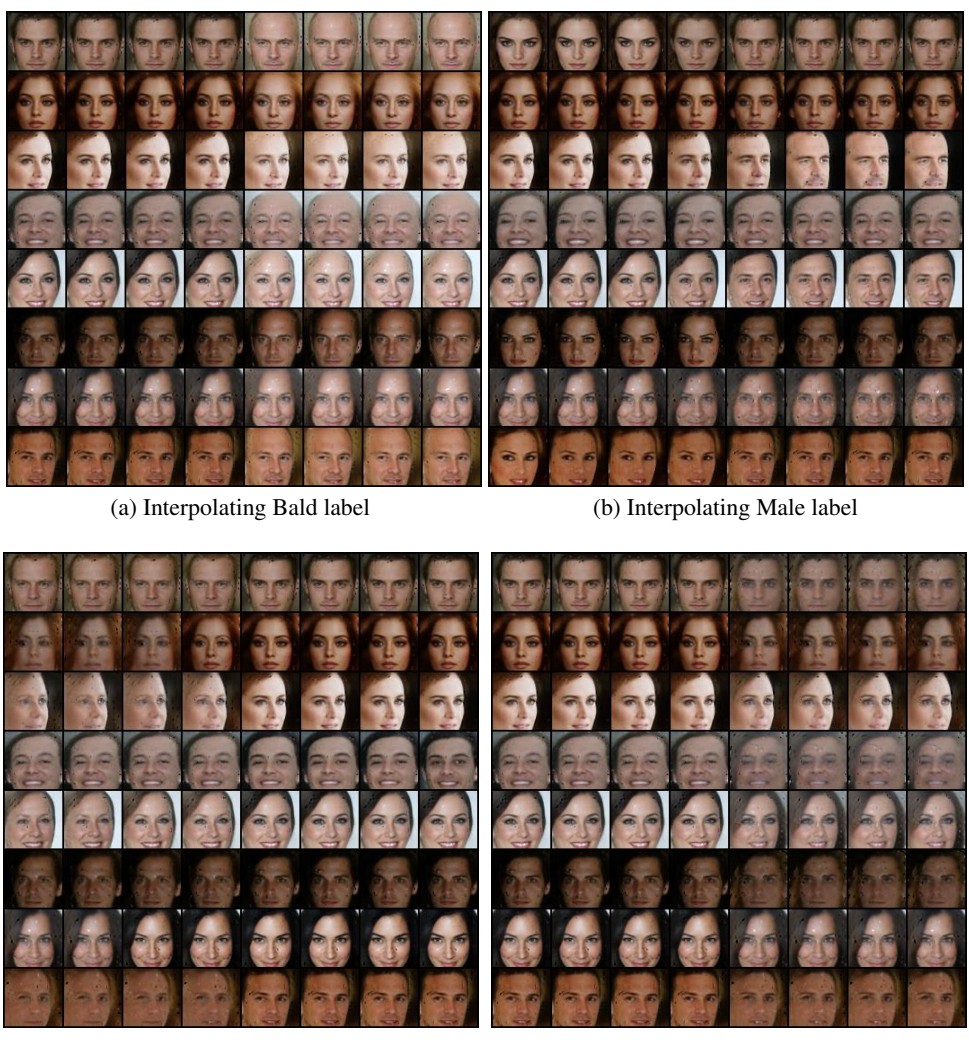

(a) Interpolating Bald label
(b) Interpolating Male label

(c) Interpolating Young label
(d) Interpolating Eyeglasses label

Figure 20: The effect of interpolating a single label for CausalBEGAN, while keeping the noise terms and other labels fixed. Although most labels are properly captured, we see that eyeglasses label is not.

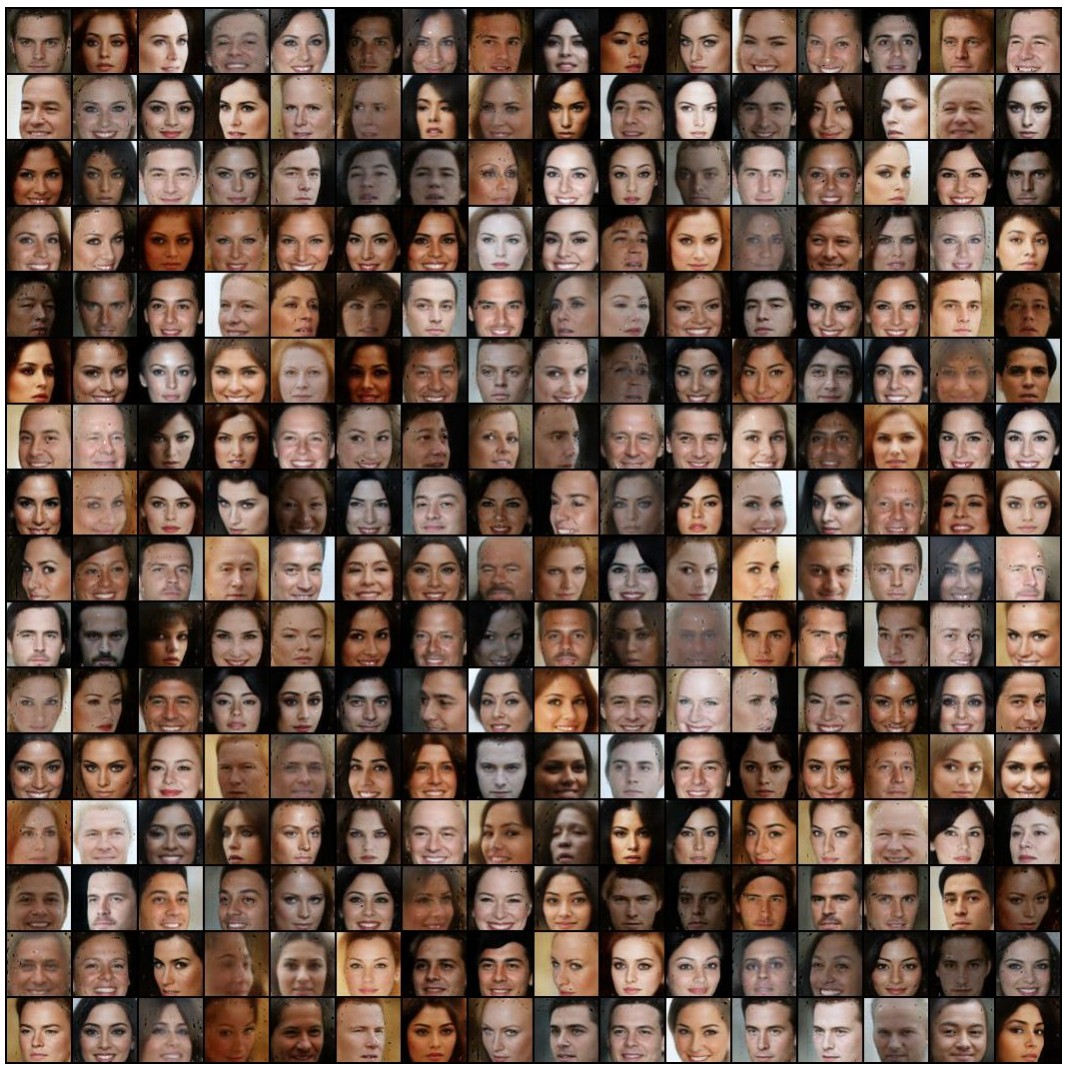

Figure 21: Diversity of Causal BEGAN showcased with 256 samples.

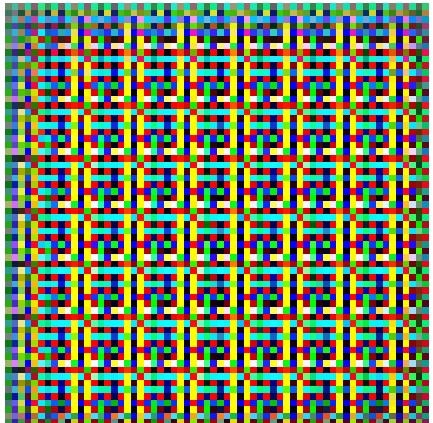

Figure 22: Failed Image generation for simultaneous label and image generation after 20k steps.

### 8.16 DIRECTLY TRAINING CiGM FOR LABELS+IMAGE FAILS

In this section, we present the result of attempting to jointly train an implicit causal generative model for labels and the image. This approach treats the image as part of the causal graph. It is not clear how exactly to feed both labels and image to discriminator, but one way is to simply encode the label as a constant image in an additional channel. We tried this for CelebA Causal Graph and observed that the image generation is not learned (Figure 22). One hypothesis is that the discriminator focuses on labels without providing useful gradients to the image generation.

## 9 IMPLEMENTATION

In this section, we explain the differences between implementation and theory, along with other implementation details for both CausalGAN and CausalBEGAN.

### 9.1 PRETRAINING CAUSAL CONTROLLER FOR FACE LABELS

In this section, we explain the implementation details of the Wasserstein Causal Controller for generating face labels. We used the total variation distance (TVD) between the distribution of generator and data distribution as a metric to decide the success of the models.

The gradient term used as a penalty is estimated by evaluating the gradient at points interpolated between the real and fake batches. Interestingly, this Wasserstein approach gives us the opportunity to train the Causal Controller to output (almost) discrete labels (See Figure 11a). In practice though, we still found benefit in rounding them before passing them to the generator.

The generator architecture is structured in accordance with Section 4 based on the causal graph in Figure 8, using uniform noise as exogenous variables and 6 layer neural networks as functions mapping parents to children. For the training, we used 25 Wasserstein discriminator (critic) updates per generator update, with a learning rate of 0.0008.

### 9.2 IMPLEMENTATION DETAILS FOR CAUSALGAN

In practice, we use stochastic gradient descent to train our model. We use *DCGAN* Radford et al. (2015), a convolutional neural net-based implementation of generative adversarial networks, and extend it into our Causal GAN framework. We have expanded it by adding our Labeler networks, training a Causal Controller network and modifying the loss functions appropriately. Compared to DCGAN an important distinction is that we make 6 generator updates for each discriminator update on average. The discriminator and labeler networks are concurrently updated in a single iteration.

Notice that the loss terms defined in Section 5.2.1 contain a single binary label. In practice we feed a $d$-dimensional label vector and need a corresponding loss function. We extend the Labeler and

Anti-Labeler loss terms by simply averaging the loss terms for every label. The $i^{th}$ coordinates of the $d$-dimensional vectors given by the labelers determine the loss terms for label $i$. Note that this is different than the architecture given in Section 8.6, where the discriminator outputs a length-$2^d$ vector and estimates the probabilities of all label combinations given the image. Therefore this approach does not have the guarantee to sample from the class conditional distributions, if the data distribution is not restricted. However, for the type of labeled image dataset we use in this work, where labels seem to be completely determined given an image, this architecture is sufficient to have the same guarantees. For the details, please see Section 8.7 in the supplementary material.

Compared to the theory we have, another difference in the implementation is that we have swapped the order of the terms in the cross entropy expressions for labeler losses. This has provided sharper images at the end of the training.

## 9.3    CONDITIONAL IMAGE GENERATION FOR CAUSALBEGAN

The labels input to CausalBEGAN are taken from the Causal Controller. We use very few parameter tunings. We use the same learning rate (0.00008) for both the generator and discriminator and do 1 update of each simultaneously (calculating the for each before applying either). We simply use $\gamma_1 = \gamma_2 = \gamma_3 = 0.5$. We do not expect the model to be very sensitive to these parameter values, as we achieve good performance without hyperparameter tweaking. We do use customized margin learning rates $\lambda_1 = 0.001, \lambda_2 = 0.00008, \lambda_3 = 0.01$, which reflect the asymmetry in how quickly the generator can respond to each margin. For example $c_2$ can have much more "spiky", fast responding behavior compared to others even when paired with a smaller learning rate, although we have not explored this parameter space in depth. In these margin behaviors, we observe that the best performing models have all three margins "active": near 0 while frequently taking small positive values.

## 9.4    ROLE OF ANTI-LABELER

In this section, we show results that compare the CausalGAN behavior with and without Anti-Labeler network. In general, using Anti-Labeler allows for faster convergence. For very rare labels, the model with Anti-Labeler provides more diverse images. See Figures 23, 24, 25.

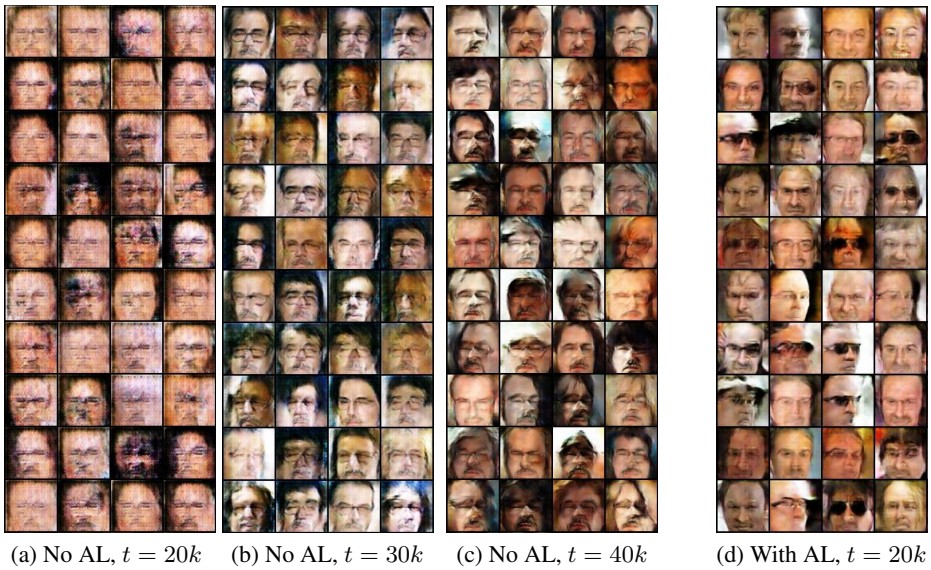

(a) No AL, $t = 20k$ (b) No AL, $t = 30k$ (c) No AL, $t = 40k$ (d) With AL, $t = 20k$

Figure 23: CausalGAN results with and without Anti-Labeler for the rare label combination *Old males with eyeglasses and mustache and narrow eyes who are not smiling*. (a, b, c) Samples without Anti-Labeler at iterations $20k, 30k, 40k$ respectively. (d) Samples with Anti-Labeler at iteration $20k$. Comparing (a) and (d), we observe that using Anti-Labeler allows for faster convergence. Comparing (c) and (d), we observe that using Anti-Labeler provides more diverse images.

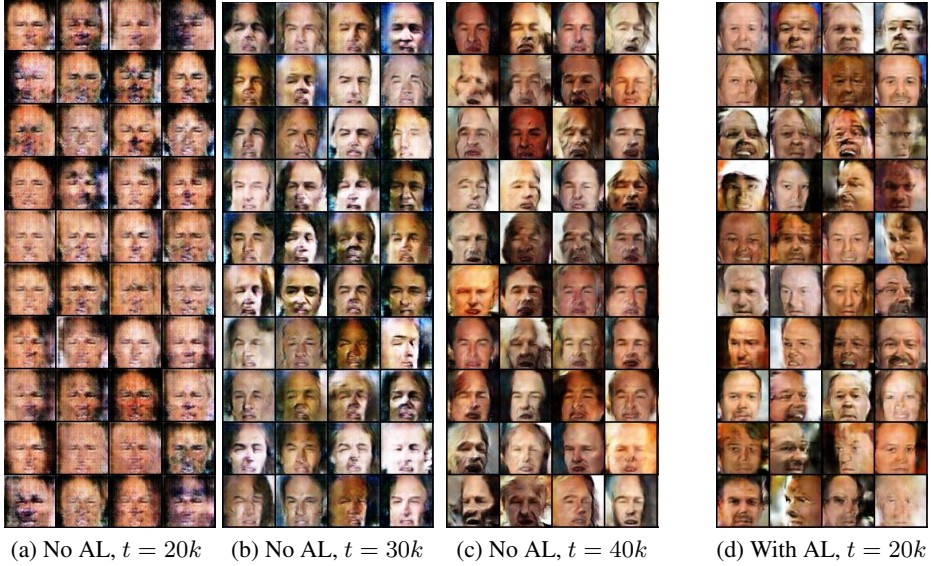

(a) No AL, $t = 20k$ (b) No AL, $t = 30k$ (c) No AL, $t = 40k$ (d) With AL, $t = 20k$

Figure 24: CausalGAN results with and without Anti-Labeler for the rare label combination *Old bald males who are not smiling but have an open mouth and narrow eyes*. (a, b, c) Samples without Anti-Labeler at iterations $20k, 30k, 40k$ respectively. (d) Samples with Anti-Labeler at iteration $20k$. Comparing (a) and (d), we observe that using Anti-Labeler allows for faster convergence.

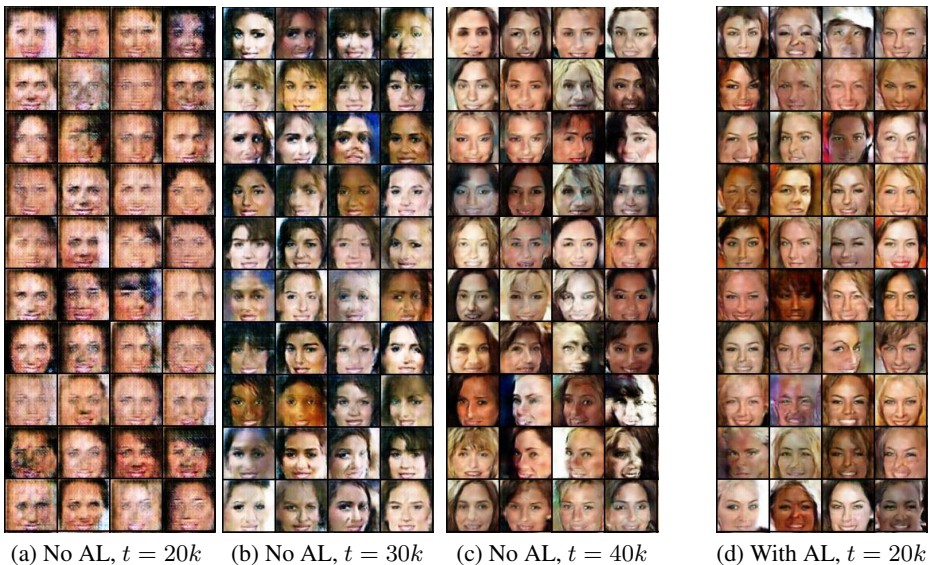

(a) No AL, $t = 20k$     (b) No AL, $t = 30k$     (c) No AL, $t = 40k$     (d) With AL, $t = 20k$

Figure 25: CausalGAN results with and without Anti-Labeler for the common label combination *Young smiling women with lipstick*. (a, b, c) Samples without Anti-Labeler at iterations $20k, 30k, 40k$ respectively. (d) Samples with Anti-Labeler at iteration $20k$. Comparing (a) and (d), we observe that using Anti-Labeler allows for faster convergence.

