# OpenReview forum: "CausalGAN: Learning Causal Implicit Generative Models with Adversarial Training"
_ICLR.cc/2018/Conference — Accept (Poster)_

### Official Review · AnonReviewer2 · 2017-11-12
**nice idea, presentation can be improved**

**Rating:** 7
**Confidence:** 3

**Review:**

In their paper "CausalGAN: Learning Causal implicit Generative Models with adv. training" the authors address the following issue: Given a causal structure between "labels" of an image (e.g. gender, mustache, smiling, etc.), one tries to learn a causal model between these variables and the image itself from observational data. Here, the image is considered to be an effect of all the labels. Such a causal model allows us to not only sample from conditional observational distributions, but also from intervention distributions. These tasks are clearly different, as nicely shown by the authors' example of "do(mustache = 1)" versus "given mustache = 1" (a sample from the latter distribution contains only men). The paper does not aim at learning causal structure from data (as clearly stated by the authors). The example images look convincing to me.

I like the idea of this paper. IMO, it is a very nice, clean, and useful approach of combining causality and the expressive power of neural networks. The paper has the potential of conveying the message of causality into the ICLR community and thereby trigger other ideas in that area. For me, it is not easy to judge the novelty of the approach, but the authors list related works, none of which seems to solve the same task. The presentation of the paper, however, should be improved significantly before publication. (In fact, because of the presentation of the paper, I was hesitating whether I should suggest acceptance.) Below, I give some examples (and suggest improvements), but there are many others. There is a risk that in its current state the paper will not generate much impact, and that would be a pity. I would therefore like to ask the authors to put a lot of effort into improving the presentation of the paper.


- I believe that I understand the authors' intention of the caption of Fig. 1, but "samples outside the dataset" is a misleading formulation. Any reasonable model does more than just reproducing the data points. I find the argumentation the authors give in Figure 6 much sharper. Even better: add the expression "P(male = 1 | mustache = 1) = 1". Then, the difference is crystal clear.
- The difference between Figures 1, 4, and 6 could be clarified.
- The list of "prior work on learning causal graphs" seems a bit random. I would add Spirtes et al 2000, Heckermann et al 1999, Peters et al 2016, and Chickering et al 2002.
- Male -> Bald does not make much sense causally (it should be Gender -> Baldness)... Aha, now I understand: The authors seem to switch between "Gender" and "Male" being random variables. Make this consistent, please.
- There are many typos and comma mistakes.
- I would introduce the do-notation much earlier. The paragraph on p. 2 is now written without do-notation ("intervening Mustache = 1 would not change the distribution"). But this way, the statements are at least very confusing (which one is "the distribution"?).
- I would get rid of the concept of CiGM. To me, it seems that this is a causal model with a neural network (NN) modeling the functions that appear in the SCM. This means, it's "just" using NNs as a model class. Instead, one could just say that one wants to learn a causal model and the proposed procedure is called CausalGAN? (This would also clarify the paper's contribution.)
- many realizations = one sample (not samples), I think.
- Fig 1: which model is used to generate the conditional sample?
- The notation changes between E and N and Z for the noises. I believe that N is supposed to be the noise in the SCM, but then maybe it should not be called E at the beginning.
- I believe Prop 1 (as it is stated) is wrong. For a reference, see Peters, Janzing, Scholkopf: Elements of Causal Inference: Foundations and Learning Algorithms (available as pdf), Definition 6.32. One requires the strict positivity of the densities (to properly define conditionals). Also, I believe the Z should be a vector, not a set.
- Below eq. (1), I am not sure what the V in P_V refers to.
- The concept of data probability density function seems weird to me. Either it is referring to the fitted model, then it's a bad name, or it's an empirical distribution, then there is no pdf, but a pmf.
- Many subscripts are used without explanation. r -> real? g -> generating? G -> generating? Sometimes, no subscripts are used (e.g., Fig 4 or figures in Sec. 8.13)
- I would get rid of Theorem 1 and explain it in words for the following reasons. (1) What is an "informal" theorem? (2) It refers to equations appearing much later. (3) It is stated again later as Theorem 2.
- Also: the name P_g does not appear anywhere else in the theorem, I think.
- Furthermore, I would reformulate the theorem. The main point is that the intervention distributions are correct (this fact seems to be there, but is "hidden" in the CIGN notation in the corollary).
- Re. the formulation in Thm 2: is it clear that there is a unique global optimum (my intuition would say there could be several), thus: better write "_a_ global minimum"?
- Fig. 3 was not very clear to me. I suggest to put more information into its caption.
- In particular, why is the dataset not used for the causal controller? I thought, that it should model the joint (empirical) distribution over the labels, and this is part of the dataset. Am I missing sth?
- IMO, the structure of the paper can be improved. Currently, Section 3 is called "Background" which does not say much. Section 4 contains CIGMs, Section 5 Causal GANs, 5.1. Causal Controller, 5.2. CausalGAN, 5.2.1. Architecture (which the causal controller is part of) etc. An alternative could be:
Sec 1: Introduction
Sec 1.1: Related Work
Sec 2: Causal Models
Sec 2.1: Causal Models using Generative Models (old: CIGM)
Sec 3: Causal GANs
Sec 3.1: Architecture (including controller)
Sec 3.2: loss functions
...
Sec 4: Empricial Results (old: Sec. 6: Results)
- "Causal Graph 1" is not a proper reference (it's Fig 23 I guess). Also, it is quite important for the paper, I think it should be in the main part.
- There are different references to the "Appendix", "Suppl. Material", or "Sec. 8" -- please be consistent (and try to avoid ambiguity by being more specific -- the appendix contains ~20 pages). Have I missed the reference to the proof of Thm 2?
- 8.1. contains copy-paste from the main text.
- "proposition from Goodfellow" -> please be more precise
- What is Fig 8 used for? Is it not sufficient to have and discuss Fig 23?
- IMO, Section 5.3. should be rewritten (also, maybe include another reference for BEGAN).
- There is a reference to Lemma 15. However, I have not found that lemma.
- I think it's quite interesting that the framework seems to also allow answering counterfactual questions for realizations that have been sampled from the model, see Fig 16. This is the case since for the generated realizations, the noise values are known. The authors may think about including a comment on that issue.
- Since this paper's main proposal is a methodological one, I would make the publication conditional on the fact that code is released.

---

> ### Author Response · Authors · 2017-12-19
> **Response to Reviewer2**
>
> Thank you for your detailed and insightful comments.
>
> - On structural changes, suggestions: Thank you for taking time to point out these points. We will add the listed references and implement all the suggested changes to make the presentation more clear, to make the wording more consistent, to fix typos, and to remove the CiGM concept, explaining it in words.
>
> - On Fig. 1: Our CausalBEGAN implementation is used to generate this figure.
>
> - On Prop. 1: Thank you for pointing this out. As you correctly observe, strict positivity of the densities is required for this to be true. We will move our assumption that label distribution is strictly positive to here as an assumption for the theorem.
>
> - On data probability density: The data probability density function corresponds to a hypothetical distribution from which the finite sized dataset was sampled.
>
> - On Theorem 1 (Informal): We will remove the “informal theorem” and replace with the a statement in words.
>
> - On formulation in Theorem 2: Since the optimization is assumed to have been done on the pdf level, and since KL divergence is zero if and only if the distributions are the same, the global minimum is unique, although there may be multiple parameterizations of the network that achieves this global minimum.
>
> - On dataset not being used for causal controller: We haven’t shown the connection to the dataset in the CausalGAN architecture figure since we assume it is already pretrained with the same dataset. We will clarify this in the figure caption.
>
> - On counterfactual samples from distribution: Thank you for your insightful comment. We do not assume that the noise distributions are known (this is not required for interventional samples to be correct). We will add a paragraph explaining that if the noise terms are known, we can use our framework to take counterfactual samples.
>
> - On code availability: The code will be made public and linked in the paper in the camera ready version.

---

### Official Review · AnonReviewer1 · 2017-11-25
**Introducing the causal mechanism into the GAN**

**Rating:** 9
**Confidence:** 3

**Review:**

This should be the first work which introduces in the causal structure into the GAN, to solve the label dependency problem. The idea is interesting and insightful. The proposed method is theoretically analyzed and experimentally tested.  Two minor concerns are 1) what is the relationship between the anti-labeler and and discriminator? 2) how the tune related weight of the different objective functions.

---

> ### Author Response · Authors · 2017-12-19
> **Response to Reviewer1**
>
> Thank you for your positive comments and feedback.
>
> - The Anti-labeler estimates labels of a given generated image. The Discriminator estimates whether a given image is real or generated, which is standard in the GAN literature. Please see Section 5.2.1 and 5.2.2 for their role and importance. 2) We did not scale the different objective functions. The main reason is that the theory we have suggests no scaling is needed, which we observe in practice.

---

### Official Review · AnonReviewer3 · 2017-11-30
**Possibly interesting, but maybe also underdeveloped. Causality part refers only to labels,**

**Rating:** 6
**Confidence:** 3

**Review:**

The paper describes a way of combining a causal graph describing the dependency structure of labels with two conditional GAN architectures (causalGAN and causalBEGAN) that generate  images conditioning on the binary labels. Ideally, this type of approach should allow not only to generate images from an observational distribution of labels (e.g. P(Moustache=1)), but also from unseen interventional distributions (e.g. P(Male=0 | do(Moustache =1)).

Maybe I misunderstood something, but one big problem I have with the paper is that for a “causalGAN” approach it doesn’t seem to do much causality. The (known) causal graph is only used to model the dependencies of the labels, which the authors call the “Causal Controller”. On this graph, one can perform interventions and get a different distribution of labels from the original causal graph (e.g. a distribution of labels in which women have the same probability as men of having moustaches). Given the labels, the rest of the architecture are extensions of conditional GANs, a causalGAN with a Labeller and an Anti-Labeller (of which I’m not completely sure I understand the necessity) and an extension of a BEGAN. The results are not particularly impressive, but that is not an issue for me.

Moreover sometimes the descriptions are a bit imprecise and unstructured. For example, Theorem 1 is more like a list of desiderata and it already contains a forward reference to page 7. The definition of intervention in the Background applies only to do-interventions (Pearl 2009) and not to general interventions (e.g. consider soft, uncertain or fat-hand interventions).

Overall, I think the paper proposes some interesting ideas, but it doesn’t explore them yet in detail. I would be interested to know what happens if the causal graph is not known, and even worse cannot be completely identified from data (so there is an equivalence class of possible graphs), or potentially is influenced by latent factors. Moreover, I would be very curious about ways to better integrate causality and generative models, that don’t focus only on the label space.


Minor details:
Personally I’m not a big fan of abusing colons (“:”) instead of points (“.”). See for example the first paragraph of the Related Work.

EDIT: I read the author's rebuttal, but it has not completely addressed my concerns, so my rating has not changed.

---

> ### Author Response · Authors · 2017-12-19
> **Response to Reviewer3**
>
> Thank you for your comments and feedback.
>
> - On the use of causality:
>
> You are correct, the causal controller is the causal part of our paper. The point is that we assume the causal graph structure but not the functions that determine the structural equations. The novelty is that the structural equations can be modeled with neural networks and learned through adversarial training.
>
> The second novelty is that by creating the image conditional GAN (with a labeler and anti-labeler), we can provably guarantee that we sample from conditional and interventional distributions of labels and images. The complexity of having a labeler and an anti-labeler is needed for our proof.
>
> Another interesting byproduct of our method is that the image generation (which is essentially a conditional GAN) can be creative, i.e., produce images that never appear in the training set which does not happen for other conditional GANs.
>
> - On structural suggestions: Thank you for your comments on structuring and presentation. Among other changes, we will remove the “informal theorem” and replace with the a statement in words.
>
> - When the causal graph is unknown:
>
> It is indeed very interesting to extend our framework for learning the causal graph structure or when there are latent variables.
> We investigate the effect of using the wrong causal graph in the appendix of the paper. We see that, as long as CIs in the data are respected, a wrong causal graph can also be learned with a GAN. As it is evident from this observation, it is not trivial to infer causality from how well the data can be fit. This is an interesting direction for future work.

---

### Public Comment · ~yoav_shalev1 · 2017-11-22
**typo?**

seems that on equations 2 and 3 (page 7) you need to switch between the l=0 and l=1 positions....

---

> ### Author Response · Authors · 2017-11-29
> **Thank you for your comment**
>
> As you correctly observe, the positions of l=0 and l=1 should be swapped in (2) and (3).

---

### Public Comment · (anonymous) · 2018-02-01
**difficult reading**

Nice and interesting paper. However, it has been a bit difficult to follow it. I agree completely with the comments of reviewer 2 and I am not sure the authors really addressed all his/her points. Also, I really don't see how reviewer 1 can only have two minor concerns (note that I have read the revised paper, which it is supposed to be an improved version of the one he/she read).

Below some specific examples of what I mean by "difficult to follow":

1. Figs. 4-7 are shown without too much explanation (for instance, what's the difference between Fig. 4 and Fig.6?).
2. The structure of the paper seems weird (why the "Theoretical Guaranties" are in section 5.2.3? I would put them earlier or in the appendix (which actually would leave some room for a more detailed results section)).
3. It took me a while to find Causal Graph 1 (Fig. 23!), which seems to be the graph used for all figures in the results section (Fig. 23 is never mentioned in that section).
4. The concept of Causal Implicit Generative Model (CiGM) is never formally explained, the best/only definition is given in the abstract.

Hope this help.

Thanks

---

> ### Author Response · Authors · 2018-02-23
> **Uploaded camera ready**
>
> Dear Anonymous,
>
>  Thank you for your positive remarks and constructive feedback. We just uploaded our camera ready version after implementing your and the reviewers' comments. We would greatly appreciate any additional feedback you may give.
>
> Regards,

---

### Decision · Program_Chairs · 2018-01-29
**ICLR 2018 Conference Acceptance Decision**

**Decision:**

Accept (Poster)

**Comment:**

This paper proposes an interesting machinery around Generative Adversarial Networks to enable sampling not only from conditional observational distributions but also from interven­tional distributions. This is an important contribution as this means that we can obtain samples with desired properties that may not be present in the training set; useful in applications such as ones involving fairness and also when data collection is expensive and biased. The main component called the causal controller models the label dependencies and drives the standard conditional GAN. As reviewers point out, the causal controller assumes the knowledge of the causal graph which is a limitation as this is not known a priori in many applications. Nevertheless, this is a strong paper that convincingly demonstrates a novel approach to incorporate causal structure into generative models. This should be of great interest to the community and may lead to interesting applications that exploit causality. I recommend acceptance.